# A single-parasite transcriptional atlas of *Toxoplasma Gondii* reveals novel control of antigen expression

Yuan Xue[1], Terence C Theisen[2], Suchita Rastogi[2], Abel Ferrel[2], Stephen R Quake[1,3,4]*, John C Boothroyd[2]*

[1]Department of Bioengineering, Stanford University, Stanford, United States; [2]Department of Microbiology and Immunology, Stanford University School of Medicine, Stanford, United States; [3]Department of Applied Physics, Stanford University, Stanford, United States; [4]Chan Zuckerberg Biohub, San Francisco, United States

**Abstract** *Toxoplasma gondii,* a protozoan parasite, undergoes a complex and poorly understood developmental process that is critical for establishing a chronic infection in its intermediate hosts. Here, we applied single-cell RNA-sequencing (scRNA-seq) on >5,400 Toxoplasma in both tachyzoite and bradyzoite stages using three widely studied strains to construct a comprehensive atlas of cell-cycle and asexual development, revealing hidden states and transcriptional factors associated with each developmental stage. Analysis of SAG1-related sequence (SRS) antigenic repertoire reveals a highly heterogeneous, sporadic expression pattern unexplained by measurement noise, cell cycle, or asexual development. Furthermore, we identified AP2IX-1 as a transcription factor that controls the switching from the ubiquitous SAG1 to rare surface antigens not previously observed in tachyzoites. In addition, comparative analysis between *Toxoplasma* and *Plasmodium* scRNA-seq results reveals concerted expression of gene sets, despite fundamental differences in cell division. Lastly, we built an interactive data-browser for visualization of our atlas resource.

*For correspondence:
steve@quake-lab.org (SRQ);
jboothr@stanford.edu (JCB)

**Competing interests:** The authors declare that no competing interests exist.

## Introduction

*Toxoplasma gondii* is an intracellular protozoan parasite that is thought to infect over a quarter of the world's population (*Pappas et al., 2009*). Like some of its *Apicomplexan* cousins, *Toxoplasma* undergoes a complex developmental transition inside the host. In intermediate hosts, including humans and virtually all other non-feline, warm-blooded animals, *Toxoplasma* parasites remain haploid and transition from a replicative, virulent tachyzoite to an encysted, quasi-dormant bradyzoite. This asexual developmental transition is tightly coupled to the clinical progression of *Toxoplasma* infection. Although acute infection with tachyzoites produces few if any symptoms in healthy human children and adults, infected individuals, if left untreated, progress to a chronic stage wherein tachyzoites transition to bradyzoites that can persist for life in neurons and muscle cells. When infected individuals become immunocompromised, such as in chemotherapy, HIV infection, or organ transplantation (*Rabaud et al., 1994*; *Robert-Gangneux et al., 2015*), bradyzoites can reactivate to become tachyzoites, causing severe neurological damage and even death. While no causal link has been established, a population-wide study has uncovered significant association of *Toxoplasma* infection with schizophrenia in chronically infected humans (*Sutterland et al., 2015*). Chronic infection in mice has been observed to induce behavioral changes such as loss of aversion to cat urine, which is hypothesized to increase the transmission rate of *Toxoplasma* to its definitive feline host where sexual reproduction occurs (*Vyas et al., 2007*). As there are no therapeutic interventions to

**eLife digest** *Toxoplasma gondii* is a single-celled parasite that can infect most warm-blooded animals, but only reproduces sexually in domestic and wild cats. Distantly related to the malaria agent, it currently infects over a quarter of the world's human population. Although it is benign in most cases, the condition can still be dangerous for foetuses and people whose immune system is compromised.

In the human body, *Toxoplasma* cells infiltrate muscle and nerve cells; there it undergoes a complex transformation that helps the parasites to stop dividing quickly and instead hide from the immune system in a dormant state. It is still unclear how this transition unfolds, and in particular which genes are switched on and off at any given time.

To understand this transformation, scientists often measure which genes are active across a group of parasites. However, this approach gives only an 'average' picture and does not allow each parasite to be profiled, missing out on the diversity that may exist between individuals. One area of particular interest, for example, is a set of genes called SAG1-related sequences. They code for the 'molecular overcoat' of the parasite, an array of proteins that sit on the surface of *Toxoplasma* cells. More than 120 SAG1-related genes exist in the genome of each *Toxoplasma* parasite, creating a whole wardrobe of proteins that potentially hide the parasites from the immune system.

Here, Xue et al. harnessed a technique called single-cell RNA sequencing, which allowed them to screen which genes were active in 5,400 individual *Toxoplasma* parasites from different strains. The analysis included both the rapidly dividing form of the parasite (present in the initial stage of an infection), and the slowly dividing form found in people who carry *Toxoplasma* without any symptoms. The resulting 'atlas' contains previously hidden information about the genes used at each stage of parasite development: this included unexpected similarities between *Toxoplasma* and the malaria agent, as well as subtle differences between two of the *Toxoplasma* strains.

The atlas also sheds light on how individual parasites turns on SAG1-related sequences. It reveals a surprising diversity in the composition of the protein coats sported by *Toxoplasma* cells at the same developmental stage, a strategy that may help to thwart the immune system. One individual parasite in particular had an unusual combination of coat and other proteins found in both the fast and slow-dividing human forms. This parasite had been grown in human cells, yet a closer analysis revealed that it had activated several genes (including ones encoding the protein coat) that are normally only 'on' in the parasites going through sexual reproduction in domestic and wild cats.

This new data atlas helps to understand how *Toxoplasma* are transmitted to and grow within humans, which could aid the development of treatments. Ultimately, a better knowledge of these parasites could also bring new information about the agent that causes malaria.

prevent or clear cysts in infected individuals, understanding how *Toxoplasma* transitions through its life stages remains of critical importance.

The development of in vitro methods to induce *Toxoplasma* differentiation have facilitated investigation of several aspects of chronic infection, including transition of tachyzoites to bradyzoites (*Soête et al., 1994*; *Jeffers et al., 2018*). Bulk transcriptomic analyses of *Toxoplasma gondii* at distinct asexual stages reveal genetic modules that are expressed in each stage (*Buchholz et al., 2011*; *Manger et al., 1998a*; *Pittman et al., 2014*; *Yip, 2007*; *Cleary et al., 2002*; *Radke et al., 2005*; *Chen et al., 2018*; *Fouts and Boothroyd, 2007*), including AP2 transcription factors that are thought to play a role in differentiation (*Hong et al., 2017*; *White et al., 2014*); however, transitioning parasites convert to the bradyzoite stage asynchronously and display a high degree of heterogeneity along the developmental pathway and in gene expression (*Soete et al., 1993*; *Watts et al., 2015*). Furthermore, parasites within the same tissue cysts have been shown to display heterogeneity in the expression of bradyzoite marker proteins (*Ferguson et al., 1994*). The transition of tachyzoites to the bradyzoite stage results in an overwhelming majority of mature bradyzoites in the $G_1$ phase of the cell cycle that divide slowly, if at all (*Radke et al., 2003*; *Sinai et al., 2016*). Furthermore, tachyzoites exhibit slower growth kinetics immediately prior to the bradyzoite transition (*Radke et al., 2003*; *Jerome et al., 1998*). This suggests that parasites exit the cell cycle to differentiate into bradyzoites, a pattern consistent with developmental processes in several other eukaryotic

organisms (*Ali et al., 2011*; *Kim et al., 2010*). Dissecting these cell cycle aspects of stage conversion requires a more detailed analysis than has been possible with bulk measurement of tachyzoite or bradyzoite populations, or with the use of genetically modified parasites coupled with chemical synchronization of cell cycle progression (*Radke et al., 2005*; *Radke and White, 1998*; *Conde de Felipe et al., 2008*; *Behnke et al., 2010*). This is because the latter approaches require large quantities of synchronized parasites and can potentially introduce artificial perturbations. Furthermore, bulk measurement fails to distinguish parasite-to-parasite variation that is independent of cell cycle or known developmental processes, potentially missing the phenotypic diversity intrinsic to a population of cells.

Single-cell RNA sequencing (scRNA-seq) offers a powerful and unbiased approach to reveal the underlying heterogeneity in an asynchronous population of cells. Droplet and FACS-based approaches have already been applied towards multicellular parasites such as *Schistosoma* to reveal developmental changes within different hosts (*Wang et al., 2018*). Recently, scRNA-seq has revealed a surprising degree of heterogeneity in another apicomplexan parasite, *Plasmodium* (*Reid et al., 2018*; *Ngara et al., 2018*; *Poran et al., 2017*; *Howick et al., 2019*). Analyses derived from these single-parasite measurements uncovered rare and critical transition events in parasite development that were undetectable in bulk measurements. Combined with novel analytical tools and increase in measurement throughput, scRNA-seq can help facilitate the discovery of regulatory factors that mediate these developmental transitions in a system-wide fashion.

Here, we performed scRNA-seq to reconstruct transcriptional dynamics of asynchronous *Toxoplasma* parasites in the course of cell cycle and asexual development in vitro. Our analysis reveals the existence of hidden cell states and rare parasites that show highly unusual patterns of gene expression associated with specific transcription factors. We also discover that individual parasites vary substantially in the expression of surface antigen genes, suggesting the possibility of a novel form of antigenic variation that may play a crucial role in host immunity evasion. Importantly, we identified a single parasite from our scRNA-seq dataset that displayed an unusual expression pattern of surface antigens, leading us to identify and validate the regulatory role of a previously uncharacterized AP2 transcription factor typically associated with parasites in sexual development. Lastly, we show that despite fundamental differences in their modes of cell division, there are conserved transcriptional programs between the asexual life cycles of *Toxoplasma gondii* and *Plasmodium berghei*. Our results combined provide the first comprehensive single-cell atlas of *Toxoplasma* in the course of asexual development and help reveal that the antigenic repertoire of this parasite is much more heterogeneous than previously appreciated.

## Results

### Technical validation of single-parasite sorting and sequencing

There are more than a dozen approaches available for single-cell isolation and transcriptome amplification. Based on benchmark comparisons, Smart-seq2 generally has higher sensitivity than competing droplet-based approaches (*Svensson et al., 2017*; *Ziegenhain et al., 2017*). We reasoned that sensitive measurement is crucial in our study, given that single *Toxoplasma gondii* parasites are at least 50-fold smaller in volume than a typical mammalian cell, and thus the average parasite gene is likely expressed with much lower copy number per cell than a typical mammalian gene. For our initial studies, we used the common Type I lab strain of *Toxoplasma*, RH, grown in vitro in human foreskin fibroblasts (HFFs). Following such growth, individual tachyzoites were released by passage through a narrow-gauge needle and then purified by fluorescence activated cell sorting (FACS) into 384-well or 96-well plates. We then synthesized, amplified, and barcoded cDNA using Smart-seq2 and Illumina Nextera protocols. We reduced the reagent cost in 384-well plates effectively by ten-fold compared to the 96-well format. The sequenced reads were bioinformatically deconvolved and grouped into individual parasites for analysis using modified bcl2fastq and custom python scripts (Materials and methods). A schematic to illustrate our experimental workflow is shown in *Figure 1*.

To ensure that our workflow efficiently captures single *Toxoplasma* parasites, we mixed equal numbers of two transgenic lines of RH, one expressing GFP and the other expressing mCherry, and sorted individual parasites into a 384 well plate based on the presence of either green or red fluorescence. After Smart-seq2 amplification, we quantified the expression of GFP and mCherry mRNAs

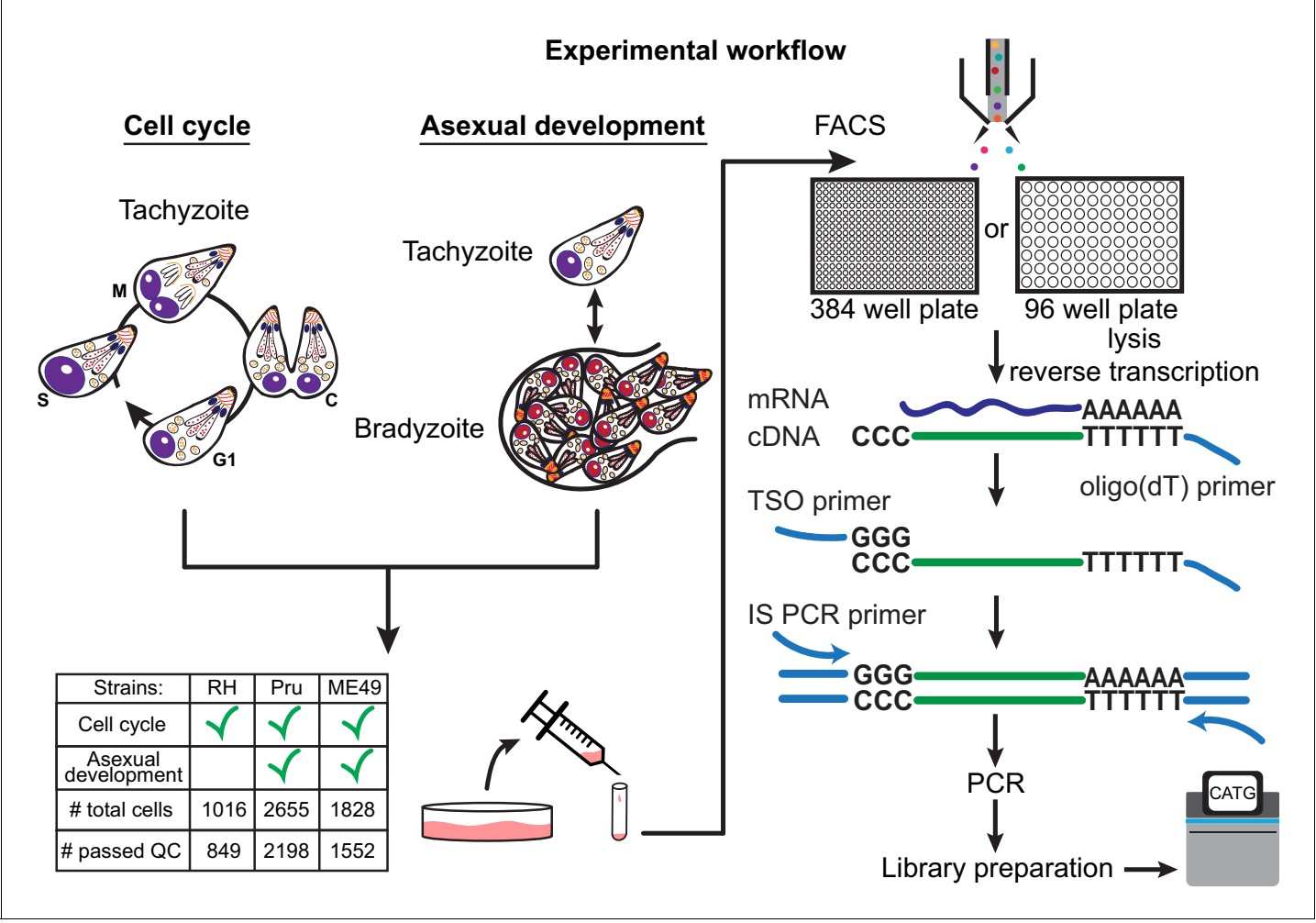

**Figure 1.** Schematic of single-cell RNA-sequencing (scRNA-seq) based on a modified Smart-seq2 protocol for 384-well or 96-well plate. A table of strain types with the number of sequenced cells and cells that passed quality checking (QC) is provided.
The online version of this article includes the following figure supplement(s) for figure 1:

**Figure supplement 1.** Technical benchmark of extracellular Toxoplasma FACS and modified gene alignment.
**Figure supplement 2.** Technical benchmark of scRNA-seq cell recovery, measurement sensitivity, and accuracy.

using quantitative polymerase chain reaction (qPCR). Across all 301 wells that we measured, we observed the presence of both GFP and mCherry mRNA in only one well, indicating that the rate of doublet events is below 1% (*Figure 1—figure supplement 1a*). To address the possibility that the reduced reagent volume in the 384-well format could potentially saturate the reaction chemistry and thus limit quantification range, we sorted varying numbers of RH and quantified with qPCR the mRNA of a gene encoding the abundantly expressed surface protein, SRS29B (SAG1) (*Figure 1—figure supplement 1b*). The detected amount of SAG1 mRNA present in wells containing single, eight, or fifty RH fell into the expected distributions based on the number of parasites sorted, without signs of saturation, indicating that the assay is capable of quantitative measurement at the single *Toxoplasma* level. We then proceeded to sort parasites into 384-well based on live/dead staining and sequence 729 RH (612 passed quality control) strain single *Toxoplasma* parasites from asynchronous populations grown under tachyzoite conditions. We also sorted and sequenced 287 RH parasites (237 passed quality control) into 96-well plate for comparison, which we will discuss further in another section. For Pru and ME49 strains, we collected parasites at several time points post alkaline treatment which induces differentiation from tachyzoites to bradyzoites to follow changes in their expression profiles during in vitro development (Materials and methods), yielding 2655 Pru (2198

passed quality control) and 1828 ME49 (1552 passed quality control) single parasites. RH reads were aligned to the GT1 strain genome, which is the most complete reference for Type I parasites, while Pru and ME49 were aligned to the ME49 strain Type II genome reference. Because many genes encoding *Toxoplasma* secretion factors and surface proteins are evolutionary products of gene duplication events (*Reid, 2015*), we expected high sequence similarity amongst a substantial portion of the parasite genes. Thus, we modified our gene counting pipeline to account for duplicated genes by distributing reads across all regions with equal alignment score that passed thresholds (Materials and methods). The reason why we adopted a correction scheme for multiply-mapped reads is because the analysis of co-occurrence, or lack thereof, of pathogenic factors (e.g. surface antigens) hinges on sensitive detection of their expression. We thus faced a choice of increasing the false positive in gene alignment by correcting for multiply mapped reads or increasing its false negative by counting only uniquely aligned reads. We chose to account for multiply mapped genes, which would otherwise be obscured, as we hypothesized that surface antigen expression may vary between individual parasites, which we further discuss in another section. A comparison of counting methods does not reveal significant differences in the observed counts (*Figure 1—figure supplement 1c*). Further analysis reveals that our modified pipeline recovered the detection of more parasite genes than default parameters (*Figure 1—figure supplement 1d*).

To ensure that poorly amplified or sequenced parasites did not confound our downstream analysis, we filtered samples based on several quality metrics including percent reads mapping to ERCC spike-in sequences, number of genes detected, and sequencing depth (Materials and methods; *Figure 1—figure supplement 2a*). On average, each sequenced parasite contains 30–50% reads that mapped to *Toxoplasma* genes encoding proteins (top panel in *Figure 1—figure supplement 2b*). Most of the unmapped reads are from *Toxoplasma*'s 28 s ribosomal RNA. The relatively high rate of rRNA contamination was also observed in single-parasite RNA sequencing of *Plasmodium* (*Reid et al., 2018*). We suspect this occurred due to non-specific priming as protozoan cells have low RNA input. We normalized for sequencing depth across cells by dividing each read count by the read sum of each corresponding cell and then multiplied by the median of read sum within each dataset to yield 'count per median' (CPM). After filtering ERCC spike-in and rRNA genes, we detected on average 862, 1290, and 970 genes per parasite with greater than or equal to two read counts (Materials and methods) in the RH, Pru, and ME49 datasets, respectively (bottom panel in *Figure 1—figure supplement 2b*). Characterization of our measurement sensitivity based on logistic regression modeling of ERCC spike-in standards (Materials and methods) (*Lönnberg et al., 2017*) reveals a 50% detection rate of 17, 17, and 26 molecules for RH, Pru, and ME49 datasets, respectively (top panels in *Figure 1—figure supplement 2c*). The sensitivity of our 384-well Smart-seq2 measurement is comparable to the previously reported range for the 96-well format (*Ziegenhain et al., 2017*). As expected from our qPCR titration experiment, scRNA-seq measurement of gene expression is quantitative at single parasite resolution based on ERCC standards. We determined that the linear dynamic range of our scRNA-seq measurement spans over three orders of magnitude (bottom panels in *Figure 1—figure supplement 2c*). Taken together, we demonstrate a scalable and cost-effective approach to measure the transcriptomic changes of individual parasites with high sensitivity and accuracy.

## Cell cycle landscape of asynchronous *Toxoplasma*

Previous work posited a potential link between bradyzoite development and cell cycle, which poses a significant challenge to the bioinformatic analysis of either process (*Radke et al., 2003*). To characterize cell cycling changes without confounding contributions from developmental processes, we first analyzed an asynchronous population of Type I RH strain parasites grown under tachyzoite conditions; this extensively passaged lab strain is known to have little propensity to switch to bradyzoites under such conditions (*Soête et al., 1994*) (Materials and methods). After filtering out genes whose expression levels did not vary significantly between individual parasites, we projected the data with principal component analysis (PCA) (Materials and methods). Interestingly, the first two principal components (PCs) reveal a circular trajectory that coincides with relative DNA content, determined using a cell permeable DNA content stain (top panel in *Figure 2a*). Unsupervised neighborhood clustering identified five distinct clusters of parasites based on their transcriptional profiles (middle panel in *Figure 2a*) (Materials and methods). We computed RNA velocity to infer transcriptional dynamics (*La Manno et al., 2018*; *Wolf et al., 2018*) and the velocity vector field indicates a

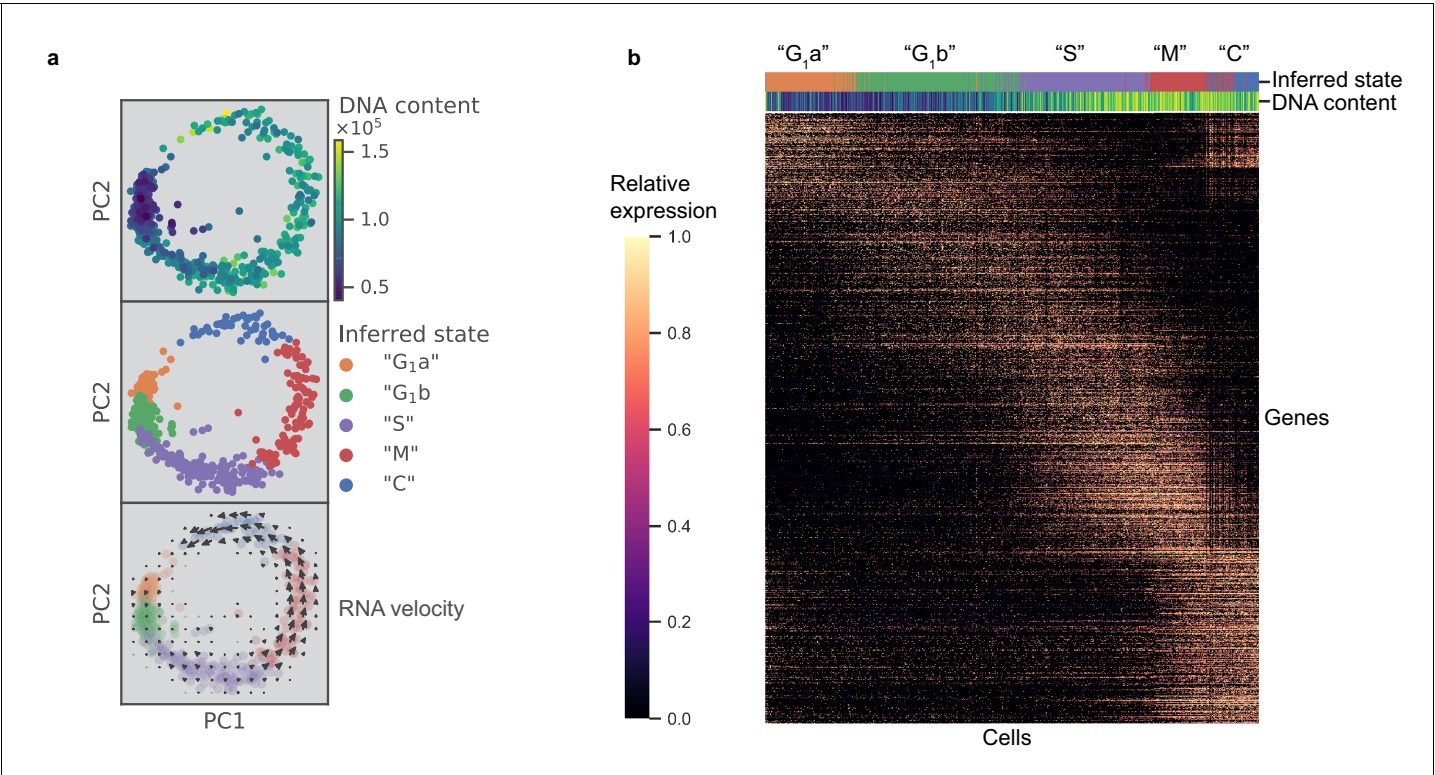

**Figure 2.** scRNA-seq resolves dynamics of Toxoplasma cell cycle in unsynchronized population. (**a**) Projection of the first two principal components in RH data set. Top panel: 612 RH cells are colored by fluorescence measurement (arbitrary unit) of a cell permeable DNA content stain, DyeCycle Violet. Center panel: cells are colored by cluster assignment and labeled by the inferred 'cell cycle' state. Bottom panel: RNA velocity vector field is overlaid on top of the inferred state colors, with arrows pointing in the direction of net transcriptional change. (**b**) Heatmap of the 1465 most variable gene expression are displayed along the rows. Cells areordered by pseudotime assignment in ascending order along the columns of the heatmap (from left to right). Top colorbar reflects the assignment of inferred state and bottom colorbar reflects the relative fluorescence of DNA content using the same color scheme as in (**a**).

The online version of this article includes the following figure supplement(s) for figure 2:

**Figure supplement 1.** Biological pseudotime analysis reveals phase-dependent expression of genetic modules.

net 'counter-clockwise' flow of transcriptional changes (bottom panel in *Figure 2a*) (Materials and methods). We assigned cell cycle phase to the clusters based primarily on change in DNA content (*Figure 2—figure supplement 1a*) but also considering previous bulk transcriptomic characterization (*Behnke et al., 2010*; *Figure 2—figure supplement 1b*). Unsupervised clustering identified two distinct clusters in $G_1$ state, which we have designated as $G_1$a and $G_1$b. We found a list of differentially expressed genes between the two $G_1$ clusters. The $G_1$a cluster is highly enriched for the expression of metabolic genes such as phenylalanine hydroxylase (*TGGT1_411100*) and pyrroline-5-carboxylate reductase (*TGGT1_236070*), as well as invasion-related secreted factors such as MIC2 (*TGGT1_201780*), MIC3 (*TGGT1_319560*), and MIC11 (*TGGT1_204530*). On the other hand, $G_1$b cluster is enriched for the expression of 3-ketoacyl reductase (*TGGT1_217740*) and cytidine and deoxycytidylate deaminase (*TGGT1_200430*), as well as numerous uncharacterized proteins (*Supplementary file 2*). The relative abundance of $G_1$a, $G_1$b, S, M, and C states were determined to be 18%, 32%, 28%, 15%, and 7%, respectively. Without chemical synchronization, the correlation between the scRNA-seq data of asynchronous parasites and previously published bulk transcriptomic measurement suggests strong agreement in cluster assignment and cell cycle state identification (*Figure 2—figure supplement 1b*). This highlights a key advantage of scRNA-seq, as it enables identification of cell cycle status of a parasite without reliance on chemical induction, which may lead to unnatural cellular behavior.

To verify the cyclical nature of gene expression through the lytic cycle, we reconstructed a biological pseudotime of RH using Monocle 2 (Materials and methods). The results show a clear oscillatory

expression pattern for the variably expressed genes along the pseudotime axis (*Figure 2b*). To further characterize cell cycle expression patterns, we clustered genes based on pseudotime interpolation and hierarchical clustering (Materials and methods). Some of the key organelles in tachyzoites are made at different times in the cell cycle (*Behnke et al., 2010*). To confirm and refine this finding, we calculated the mean expression values for each set of organelle-specific genes based on their annotation in ToxoDB (*Gajria et al., 2008*; *Supplementary file 4*). This showed the expected, strong oscillation with pseudotime (bottom panel in *Figure 2—figure supplement 1c*), which also strongly correlates with the oscillation of DNA and total mRNA content (top panels in *Figure 2—figure supplement 1c*). On the other hand, we also observed instances where a given gene's expression was discordant to the dominant trend of its nominal organelle set (arrows in *Figure 2—figure supplement 1d*). For example, 63.5% of genes annotated as rhoptry (ROP) or rhoptry neck (RON) are assigned pseudotime cluster 3, while the remaining 36.5% rhoptry genes are assigned pseudotime clusters 1 or 2 (*Figure 2—figure supplement 1e*). Specifically, genes annotated as ROP33 and ROP34, based on their homology to genes encoding known rhoptry proteins, are assigned to cluster 2 instead of cluster 3 (left panel in *Figure 2—figure supplement 1f*). Recent reports have experimentally determined these two to be non-rhoptry-localizing proteins. This is consistent with our observation of discordance between their and known ROPs' expression profiles along the pseudotime (*Beraki et al., 2019*). Through analysis of pseudotime clustering, we also identified genes not annotated as ROPs within the ROP-dominated cluster 3, such as *TGGT1_218270* and *TGGT1_230350*, that have recently been shown to encode *bona fide* rhoptry and rhoptry neck proteins, now designated as ROP48 and RON11, respectively (*Camejo et al., 2014*; *Beck et al., 2013*) (left panel in *Figure 2—figure supplement 1f*). As another example, IMC2a peaks in expression level in $G_1$, while the majority of inner-membrane complex (IMC) genes are expressed towards the M/C phase of the cell cycle (right panel in *Figure 2—figure supplement 1f*). A recent report has proposed reannotation of IMC2a as a dense granule (GRA) protein (GRA44) based on subcellular localization (*Coffey et al., 2018*), which is consistent with our unsupervised group assignment of IMC2a as falling in cluster one where GRA genes dominate. A list of 8590 RH genes with their corresponding pseudotime clustering assignment is provided (*Supplementary file 5*). We observe high discordance of pseudotime expression for several genes in each annotated organelle sets, suggesting that the current *Toxoplasma* annotation may need significant revision. Our scRNA-seq data provide an important resource to help identify mis-annotated genes and infer putative functions of uncharacterized proteins.

## Hidden heterogeneity in asexually developing *Toxoplasma*

*Toxoplasma* has one of the most complicated developmental programs of any single-celled organism; however, it is unknown how synchronized the transition is between developmental states. To address this, we assessed the inherent heterogeneity within asexually developing Pru, a type II strain that is capable of forming tissue cysts with characteristics that resemble early 'bradyzoites' from in vivo source upon growth in in vitro alkaline conditions (*Soete et al., 1993*; *Jones et al., 2017*). We applied scRNA-seq to measure and analyze Pru parasites grown in HFFs as tachyzoites ('uninduced') and after inducing the switch to bradyzoites by growth in alkaline media for 3, 5, and 7 days. Projection of the first two PCs of uninduced Pru tachyzoites (Day 0) reveals the expected circular projection (*Figure 3—figure supplement 1a*), presumably reflecting cell cycle progression as seen for the RH tachyzoites, described above. To validate this, we developed a random forest classifier model based on our cell cycle assignment in RH (Materials and methods). Comparable to what we observed in RH, cell cycle prediction reveals that the uninduced population of Pru is composed of 28%, 41%, 21%, 7%, and 3% parasites in $G_1a$, $G_1b$, S, M, and C states, respectively. Consistent with previous observation (*Jerome et al., 1998*), our data show most induced Pru parasites (Day 3–7) are in the $G_1$ state with a predominance of $G_1b$ (*Figure 3—figure supplement 1b*).

To identify transcriptomic changes associated with the tachyzoite-bradyzoite transition, we next projected data from both induced and uninduced Pru parasites onto two dimensions using UMAP, a nonlinear dimensionality reduction method (Materials and methods) (*McInnes et al., 2018*). Unsupervised clustering revealed six distinct clusters of parasites, which we label P1-6 (*Figure 3a*). Cluster formations partially correlate with treatment time points and cell cycle states (*Figure 3b*; *Figure 3—figure supplement 1c*), suggesting that the asexual differentiation program overlaps with cell cycle regulation in *Toxoplasma,* as proposed previously (*Radke et al., 2003*). We stratified the datasets by

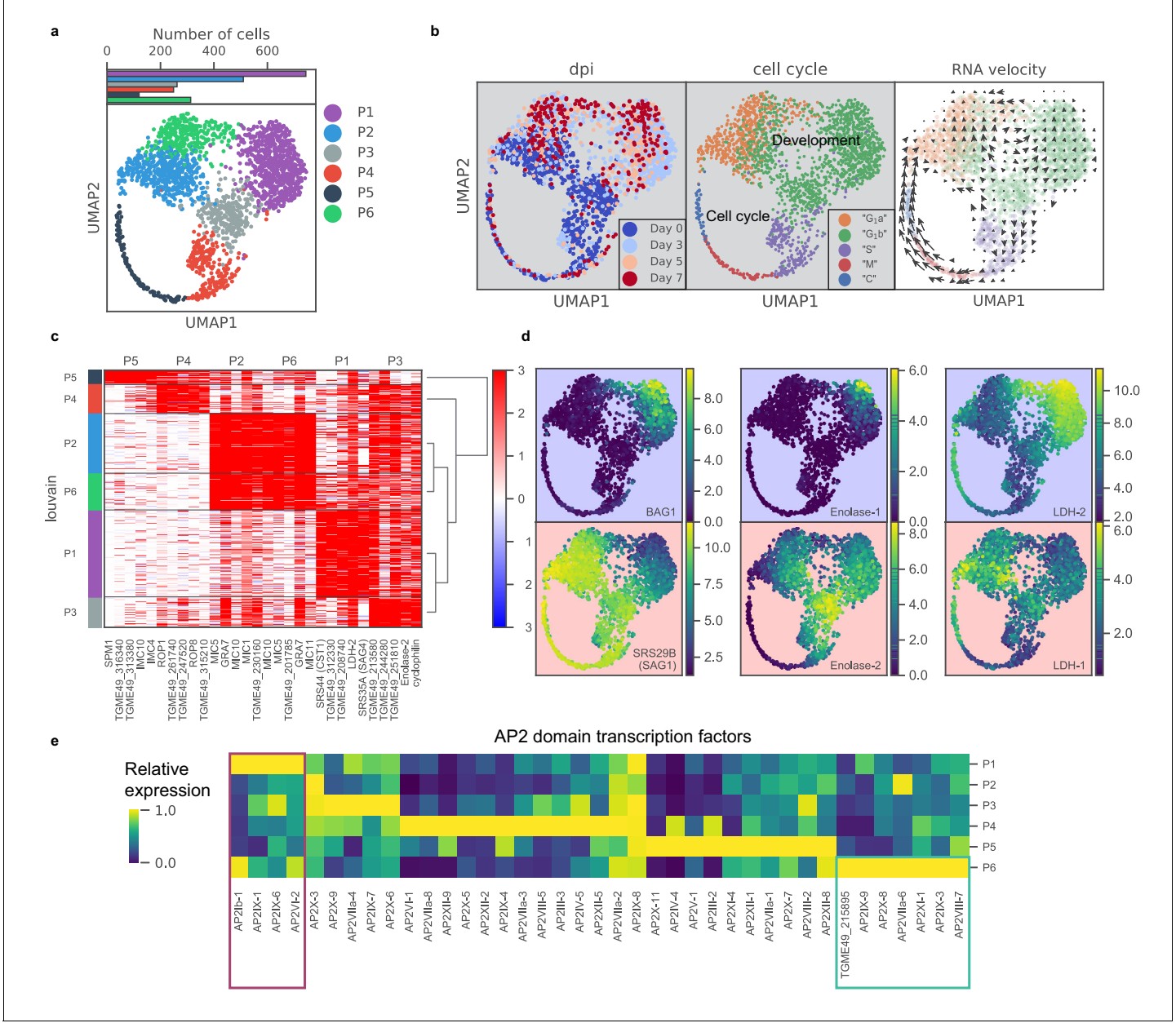

**Figure 3.** scRNA-seq resolves dynamics of asexual development in vitro. (**a**) UMAP projection of 809 uninduced and 1389 induced Pru parasites with colors indicating Louvain cluster assignment. Top panel shows the number of parasites in each cluster. (**b**) UMAP projections of Pru parasites colored or labeled by days post induction (dpi), inferred cell cycle states, and RNA velocity from left to right. (**c**) Heatmap of differentially expressed genes (along columns) across Louvain clusters of cells ordered by hierarchical clustering (along rows). The top five most enriched genes from each cluster are presented. (**d**) UMAP projections of Pru colored by the neighbor-averaged expression (log₂ CPM) of bradyzoite (top panels, purple background) and tachyzoite (bottom panels, red background) marker genes. (**e**) Heatmap of average expression level of AP2 transcription factors for each Louvain cluster, normalized by the maximum cluster expression level within each AP2. Purple and green rectangles highlight AP2s enriched in clusters P1 and P6, respectively.

The online version of this article includes the following figure supplement(s) for figure 3:

**Figure supplement 1.** Alkaline induction altered cell-cycle dynamics of Toxoplasma.

**Figure supplement 2.** Expression level (log₂ CPM) of four 'bradyzoite-specific' marker genes compared to that of 'tachyzoite-specific' marker gene, *SAG1*, stratified by days post induction (dpi; columns).

**Figure supplement 3.** Differentially expressed genes in the hidden state of alkaline-induced Pru parasites.

**Figure supplement 4.** scRNA-seq comparative analysis of asexual development in two commonly studied Toxoplasma strains.

days post alkaline induction (dpi) and observed elevated expression of previously described 'early' bradyzoite marker genes in induced parasites, including *SRS44* (*CST1*) and *BAG1*, with a concomitant reduction in expression of *SRS29B (SAG1)*, a tachyzoite-specific surface marker gene (*Figure 3—figure supplement 2*). The abundance of SAG1$^+$ parasites (72%) in the induced population suggests two possible interpretations: (1) the depletion of SAG1 mRNA is relatively slow and we are measuring SAG1 transcripts made when the parasites were still tachyzoites, or (2) the asexual transition induced by alkaline treatment is highly asynchronous. Interestingly, RNA velocity analysis suggests that P3 may be a fate decision point as the trajectory trifurcates into either P4 (cell cycle), P1, or P2 as evident by the net transcriptional flow (compare *Figure 3a* to right panel in *Figure 3b*).

To determine the gene modules specific to a given cluster, we conducted differential gene expression for each cluster (*Figure 3c* and *Supplementary file 5*). P1 cluster cells are enriched for expression of bradyzoite-specific genes while P2-5 are enriched for that of tachyzoite-specific or cell cycle-associated genes (*Figure 3c–d*). In our scRNA-seq data, we also observe a small portion of *BAG1*$^+$ bradyzoites (7.1%) annotated as either S, M, or C states, indicating that they are replicating (*Figure 3—figure supplement 1d*). Our data supports the notion that bradyzoites can undergo cell cycle progression (*Watts et al., 2015*). We observe a family of transcription factors known as Apetala 2 (AP2) that are differentially expressed across different clusters, some of which are implicated in *Toxoplasma* development (*De Silva et al., 2008*; *Radke et al., 2013*; *Walker et al., 2013*; *Figure 3e*). In particular, we identify AP2Ib-1, AP2IX-1, AP2IX-6, and AP2VI-2 as over-expressed in P1, suggesting their potential roles in the regulation of developmental transition, while AP2IX-9, AP2X-8, AP2VIIa-6, AP2XI-1, AP2IX-3, AP2VIII-7, and AP2-domain protein TGME49_215895 (not yet assigned a formal AP2 number), are highly expressed in P6, hinting at their possible roles in defining this distinct cluster of parasites.

The most highly expressed genes in P6 include genes enriched in P2 as well as bradyzoite-specific genes found in P1 (*Figure 3c*). To identify genes that are specifically expressed in P6, we used Wilcoxon's test (*Figure 3—figure supplement 3a*) (Materials and methods) between P6 and P2 or P1. Comparison of our data to previous bulk transcriptomic measurement in tachyzoites, tissue cyst, or isolates at the beginning or the end of sexual cycle showed no specific enrichment in known developmental stages (*Figure 3—figure supplement 3b*; *Ramakrishnan et al., 2019*). Instead, we show that based on their expression, P6 forms a distinct sub-population of parasites which suggests that alkaline induced *Toxoplasma* may be more heterogeneous than previously thought. Thus, scRNA-seq resolves a transcriptomic landscape of asexual development and suggests the existence of an otherwise hidden state.

To determine the reproducibility of the phenomena we observed in the differentiating Pru strain parasites, we repeated the analysis with another widely used Type II strain, ME49, examining 1828 single ME49 parasites exposed to alkaline conditions to induce switching to bradyzoites. Data from the two experiments were computationally aligned using Scanorama to remove technical batch effects while retaining sample-specific differences (*Hie et al., 2019*). Unsupervised clustering revealed five distinct clusters in ME49 which share significant overlap in expression patterns with Pru (*Figure 3—figure supplement 4a*). Matrix correlation of batch-corrected expression across the two strains demonstrate analogous mapping for most, but not all cluster identities (*Figure 3—figure supplement 4b*). To simplify the visualization and comparison across the two datasets, we next applied Partition-Based Graph Abstraction (PAGA) to present clusters of cells as nodes with connectivity based on similarity of the transcriptional profiles between clusters (*Wolf et al., 2019*). A side-by-side comparison of expression of tachyzoite, bradyzoite, and sexual stage specific genes reveals some key similarities and dissimilarities (*Figure 3—figure supplement 4c*). Clusters P1 and M1 are both enriched for the expression of bradyzoite marker genes, while clusters M4-5 and P4-5 are both predicted to be S/M/C phases of the cell cycle. Curiously, P6-specific genes (green panels in *Figure 3—figure supplement 4c*) are not enriched in any cluster in ME49, suggesting that P6 state is not conserved across the two type II strains. Such differences may not be surprising, however, as Pru and ME49 have entirely distinct passage histories.

## scRNA-seq characterizes surface antigenic repertoire of *Toxoplasma*

A unique advantage of scRNA-seq over bulk RNA-seq is its ability to measure cell-to-cell variation that is independent of known biological processes. The *Toxoplasma gondii* genome encodes a family of over 120 SAG1-related sequence (SRS) proteins that fall into distinct subfamilies; most or all of

these are presumed to be surface antigens based on their sequence similarities, including the presence of a predicted GPI-addition signal (*Manger et al., 1998b*). Whether *SRSs* constitute an antigenic repertoire that contribute to evasion of host adaptive immunity response is unclear; however, existing data on developmentally regulated expression of *SRSs* including *SAG1* (*SRS29B*) and *SRS16B* lend support to that hypothesis (*Kim and Boothroyd, 2005*; *Kim et al., 2007*). In the experiments using 384-well plates, we noted that most *SRS* genes were detected in only a few sporadic cells (*Figure 4—figure supplement 1a*); however, biological variation was difficult to distinguish from measurement dropout (failure to capture a mRNA molecule) given the limited sensitivity of 384-well assay and the relatively low abundance of *SRS* transcripts. To increase our sensitivity, therefore, we performed scRNA-seq of extracellular *Toxoplasma* in 96-well plates which achieved ~40% sensitivity of detection for single molecules of ERCC spike-ins, compared to 14% sensitivity that we obtained in the 384-well experiments, at roughly equivalent sequencing depth (*Figure 4a*). To determine the extent of transcriptional variation in *SRS* genes independent of cell cycle or asexual development, we isolated RH parasites in G1 state using DNA content stain and FACS (*Figure 4—figure supplement 1b*). We noticed that the distributions of DNA content in single cells varied across the two experiments (compare *Figure 4—figure supplement 1b* to *Figure 3—figure supplement 1a*), suggesting that the distribution may depend on external factors such as tissue culture confluence, parasite load, or the amount of time passed since infection. We also measured on average 984 genes with read counts equal to or greater than two in 96-well plates, up from 862 genes in 384-well plates (*Figure 4—figure supplement 1c*). Analysis of the resulting 96-well plates data show that the vast majority of *SRS* genes are detected in only a small fraction of the population (*Figure 4b*), similar to what we observed in 384-well experiments. To control for measurement dropout as a cause of such variation, we first determine the non-zero median for those genes, that is, the median expression level in cells where any transcript for that gene is detected. We then assess how often we fail to detect an ERCC spike-ins that had a similar non-zero median expression level. If the failure to detect a *SRS* was due to measurement dropout, then the fraction of cells without detection for its transcript should be about the same as for the ERCC spike-in with similar non-zero median expression. If, on the other hand, the failure to detect the *SRS* transcript is due to underlying biological variation between cells, then we will find a lower frequency of cells with detectable transcript for that gene than the ERCC spike-in with similar expression. The results show that *SRSs* are indeed detected at a substantially lower rate when compared to ERCC spike-ins (*Figure 4c*), indicating that the variation of *SRS* detection cannot be explained by measurement noise. Compared to most other genes, *SRSs* are expressed in a significantly smaller fraction of the population and at a relatively lower abundance (*Figure 4—figure supplement 1d-e*). To determine whether variation of *SRS* expression is due to cell cycle or asexual development, we developed a bootstrapping approach to quantify the dependence of expression on a topological network that represents either process (Materials and methods). We show that apart from a few known *SRSs*, most *SRSs* do not co-vary with cell cycle or asexual development, which suggests that the sporadic nature of *SRS* expression may hint at a different biological role for their variation, beyond cell cycle and asexual development, as discussed further below (*Figure 4d*). These results reveal new insights into the antigenic repertoire of *Toxoplasma*, which appears far more heterogeneous, on a parasite-to-parasite basis, than previously known.

## Transient expression of AP2IX-1 induces surface antigen switching

In examining our data, we were struck by a lone cell in the 384-well dataset of RH that had no detectable level of *SRS29B* (*SAG1*), which was otherwise ubiquitously and abundantly expressed in all tachyzoites (*Figure 5a*). We wondered if this cell was damaged or unhealthy but its gene count and read depth were similar to its cohort (*Figure 5b*). Further analysis of this SAG1⁻ cell's transcriptome revealed that it lacks expression of the most abundant tachyzoite-specific and bradyzoite-specific genes; instead, its gene expression most closely resembles the sexual stages from cat intestinal isolates, including abundant expression of a cat-stage *SRS*, *SRS22C* (*Figure 5c*). Within the set of genes uniquely expressed in the SAG1⁻ cell are *AP2IX-1* and *AP2III-4* which suggest a possible role for one or both of these transcription factors in regulating the transcript differences observed in this outlier. To test this hypothesis, we transiently expressed AP2IX-1 under the control of a strong promoter in RH parasites (*Figure 5d*). Consistent with our hypothesis, immunofluorescence assay (IFA) revealed a reduction of SAG1 surface protein expression within ~18–20 hr after transfection

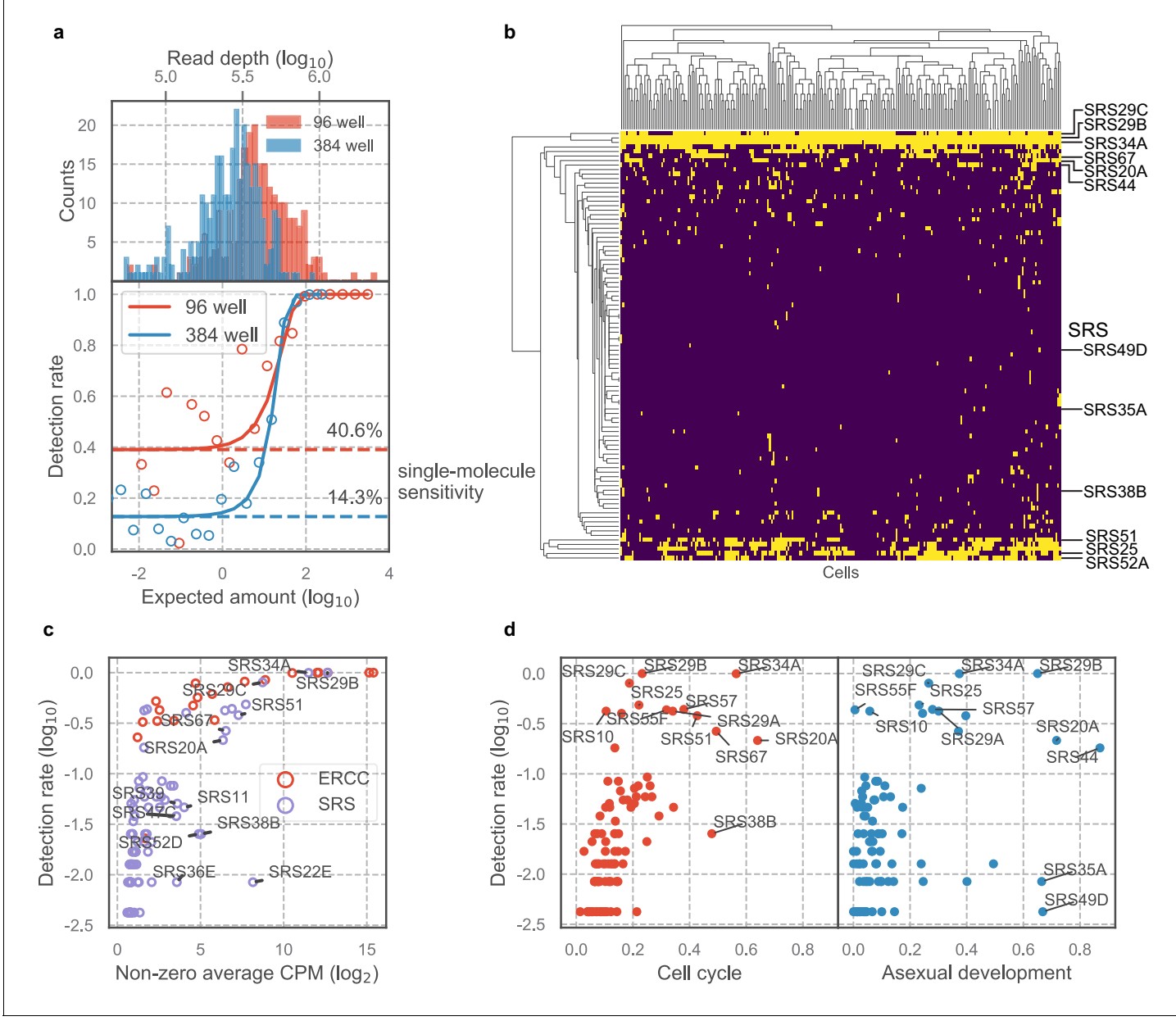

**Figure 4.** Majority of SAG1-related sequences (SRS) antigens are variably expressed in individual parasites. (**a**) Read depth (top) and detection rate of ERCC spike-ins (bottom) reveal higher measurement sensitvity in 96 well plate over 384 well plate format at roughly equivalent sequencing depth. (**b**) Hierarchical clustering heatmap of binarized expression reveals sparse expression pattern of SRS. SRS gene expression is binarized by converting expression ≥2 read counts to one and <2 to 0. (**c**) Comparison of ERCCs with similar mean expression level reveals that low detection rate of SRS cannot be fully explained by measurement dropout. (**d**) We quantified the projection dependence score of SRS expression to cell cycle projection in 384-well RH or asexual development projection in 384-well Pru, reflecting the degree of co-variation of SRS expression with respect to these two biological processes. The detection rate in 96-well RH dataset is plotted along the y-axis. This analysis reveals that some of the SRS variation cannot be readily explained by either cell cycle or asexual development.

The online version of this article includes the following figure supplement(s) for figure 4:

**Figure supplement 1.** Technical benchmark of 96-well scRNA-seq measurement.

(*Figure 5e–f*), while quantitative RT-PCR showed significantly higher mRNA expression for several of the genes that were upregulated in the SAG1⁻ cell: namely, 207965, 222305, 205210, and SRS22C, the latter being over 1000-fold higher in the transfected population than control (*Figure 5g*). Note that the SAG1 transcript levels in the transfected population were not substantially lower, as

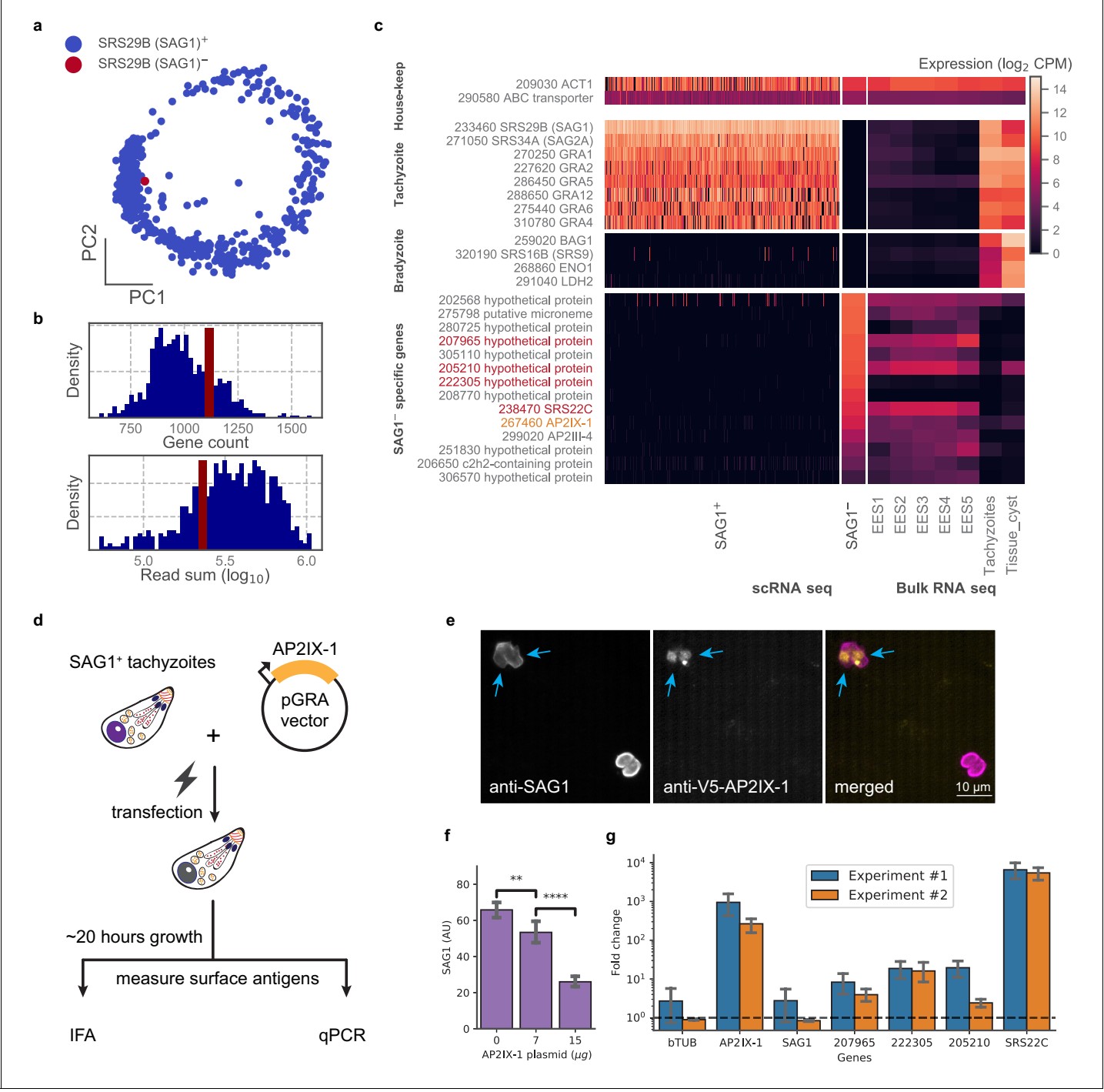

**Figure 5.** Analysis of a single SAG1⁻ outlier cell reveals regulatory role of AP2IX-1 on cat-stage antigen and genes. (**a**) PCA projection of 384-well RH cells colored by the presence (blue) or absence (red) of the SRS29B (SAG1) gene. (**b**) Gene count and total reads mapped for SAG1⁺ (blue) and SAG1⁻ (red) cells. (**c**) Comparison of house-keeping, tachyzoite-specific, bradyzoite-specific, and SAG1⁻ enriched gene expression (log₂ CPM) in scRNA-seq to bulk RNA-seq measurement of enteroepithelial stages (EES1–EES5), tachyzoites, and tissue cysts in RH. Genes that were pursued further by qRT-PCR are indicated in red; AP2IX-1 is colored orange. (**d**) Schematic illustration of transfection experiment. AP-2 IX-1 from genomic DNA of RH was cloned into a pGRA1 backbone vector and expressed constitutively in transfected parasites. Expression level of gene and protein was quantified with immunofluorescence assay (IFA) and qRT-PCR. (**e**) Representative IFA images showing anti-SAG1 (left), anti-V5-AP2IX-1 (center), and merged channels (right). Blue arrows point at parasites with detected expression of V5-AP2IX-1. (**f**) IFA quantification of anti-SAG1 signal shows dosage dependence of SAG1 protein expression on amount of AP2IX-1 plasmid transfected, with error bars depicting the standard deviation of quantification and asterisks representing statistical significance of two-tailed t-test for independence. (**g**) qPCR measurement shows that transfection of 15 µg AP2IX-1 plasmid,

*Figure 5 continued on next page*

*Figure 5 continued*

compared to no DNA control, induced the expression of AP2IX-1 and several putative surface antigens, including SRS22C, in two independent experiments with each measured in biological triplicates. AP2IX-1 transfection did not lead to increased expression of beta tubulin (bTUB) or SAG1. Error bars depict standard deviation of the measurement.

The online version of this article includes the following source data for figure 5:

**Source data 1.** Immunofluorescence assay (IFA) quantification results of SAG1 and V5 expression level in AP2IX-1 transfected population.
**Source data 2.** qPCR quantification results of AP2IX-1 transfection.

expected because there remains a large number of untransfected cells in the population which still express high levels of this gene. As *SRS22C* is predicted to be a surface antigen like the rest of *SRS* (*Gajria et al., 2008*), this suggests that AP2IX-1 induction can control the switching of surface antigens. We also attempted to express *AP2III-4* that is upregulated in the SAG1⁻ cell but were unable to obtain an epitope-tagged version of the gene (which is over nine kbp, including the promoter). Overall, these results demonstrate the ability to infer transcriptional regulation from a single parasite cell that has an unusual co-expression pattern, revealing AP2IX-1 as a novel transcriptional factor that can alter antigen expression in *Toxoplasma*.

## Comparative analysis of *Plasmodium* and *Toxoplasma* reveals shared expression programs

*Plasmodium* and *Toxoplasma* are both unusual for their prevalence in humans and their complex developmental transition, despite the fact that humans are non-definitive hosts for both pathogens. *Toxoplasma* has been used as an experimental model for other apicomplexans including *Plasmodium*, yet the replication modes amongst apicomplexans can be very different. For example, asexual replication of *Toxoplasma* involves endodyogeny, which is similar to canonical binary fission, in order to replicate and divide. Asexual division of *Plasmodium*, on the other hand, involves schizogony, a process that entails multiple rounds of DNA replication and nuclear division followed by a mass cytokinesis (*Aly et al., 2009*; *Cowman et al., 2016*). Despite their fundamental differences in cell cycle progression, we wondered if our *Toxoplasma* dataset can be combined with the Malaria Atlas (*Howick et al., 2019*) to provide insights into apicomplexan biology. Toward this end, we first identified 1830 one-to-one orthologous genes between *Plasmodium berghei* and *Toxoplasma gondii* Pru (*Figure 6—figure supplement 1a*). As expected, the vast majority of orthologous genes are not surface adhesion factors or effector molecules in *Toxoplasma* (*Figure 6—figure supplement 1b*). After removing non-orthologous genes, we combined the two species' datasets to produce an integrated UMAP projection with Scanorama (*Hie et al., 2018*). In this integrated projection space, mutually similar clusters of cells in the two datasets are brought close together, while organism-specific cell types are not (*Figure 6a*). Transitional dynamics of the parasitic development are preserved in the integrated projection, as separation of the original cluster assignment indicates (*Figure 6b*). We highlight comparative similarity across the two organisms by calculating the fraction of cells that share the same topological neighborhood in the integrated network of *Plasmodium* and *Toxoplasma* (*Figure 6c*), revealing striking similarity of expression pattern between the two organisms. We discover the concerted transcriptional expression of several conserved orthologous gene sets in the life cycles of these two parasites (*Figure 6d*). For example, the pre-erythrocytic stages of *Plasmodium* (sporozoite and merozoites) most closely match the 'G1 a' stage of *Toxoplasma*. Both parasites at these stages are not actively replicating DNA and express high levels of ribosomal and mitochondrial genes. Meanwhile, the ring stage of *Plasmodium* most closely matches the 'G1 b' phase of *Toxoplasma* because they both express high levels of ribosomal genes with minimal expression of IMC-related and microtubule-related genes. Exo-erythrocytic form (EEF), microgametes (male), and trophozoite stages of *Plasmodium berghei* most closely match the 'S' phase of *Toxoplasma* cell cycle. This is because they express DNA-replication factors and centrosome components, which are required for condensation and segregation of chromosomes. Intriguingly, the schizont stage of *Plasmodium* most closely resembles the 'M' and 'C' phases due to the expression of microtubule, centrosome, and IMC genes. A more detailed illustration of these gene sets expression is shown in *Figure 6—figure supplement 1c*. Our results indicate that while the cellular morphology and replication strategies may differ drastically between the two parasites, the cellular state, as defined by

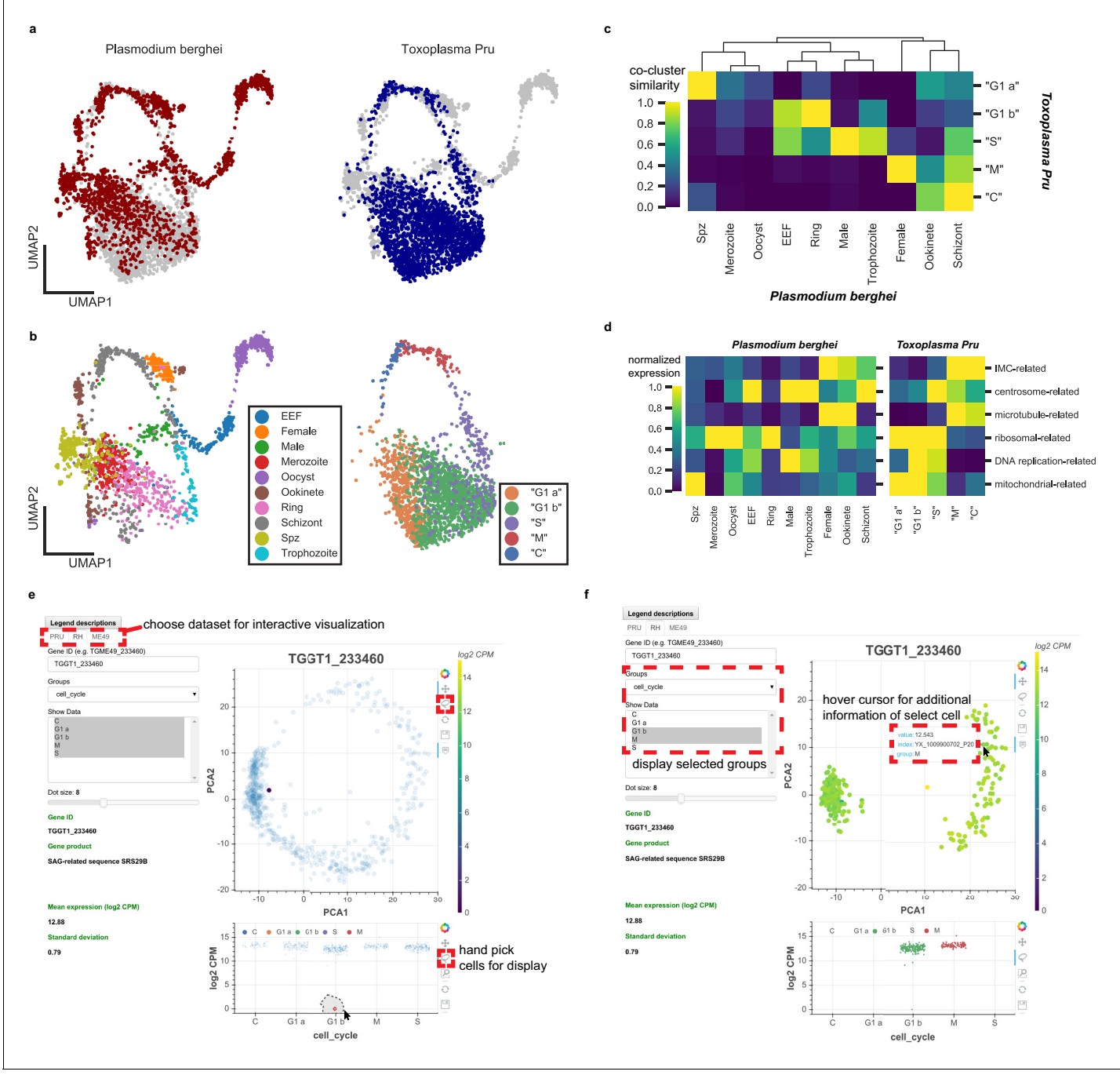

**Figure 6.** Comparative analysis of scRNA-seq in Toxoplasma gondii and Plasmodium berghei reveals concerted genetic programs underlying their life-cycles. (**a**) Scanorama integration of *Plasmodium berghei* (red, left) and *Toxoplasma gondii* Pru (blue, right). (**b**) Cell cycle of *Toxoplasma* is well-aligned to the erythrocyte cycle of *Plasmodium berghei*, despite fundamental differences in cell cycle progression between these two apicomplexans. Each cell is colored by the original cluster assignment in the corresponding dataset. (**c**) Normalized cluster similarity between the original cluster assignment of *Plasmodium berghei* and *Toxoplasma Pru*. Cluster similarity is calculated by quantifying the fraction of cells that overlap in topological network in each cluster of the corresponding dataset. (**d**) Heatmap of concerted gene sets expression normalized to one within each cluster of cells in *Plasmodium* (left) and *Toxoplasma* (right). (**e**) Single-cell atlas of *Toxoplasma gondii* can be interactively visualized (http://st-atlas.org). Individual cells with distinct expression pattern can be highlighted using built-in graphical interface tools. (**f**) Groups of cells can be selectively displayed based on cell cycle state, developmental clusters, and other categories. Hovering over an individual cell with cursor reveals gene expression level, sample id, and group membership for further analysis. All our datasets are available for download on the atlas.

The online version of this article includes the following figure supplement(s) for figure 6:

**Figure supplement 1.** Concerted expression profile of orthologous gene sets across the two distantly-related apicomplexans.

the transcriptomic profiles, can bear striking analogy and resemblance in the course of asexual replication.

## An interactive resource for visualizing single-*Toxoplasma* atlas

Toward enabling the larger scientific community to take advantage of the scRNA-seq data we collected on the three *Toxoplasma* strains, we have constructed an interactive single-cell atlas for *Toxoplasma gondii* (http://st-atlas.org) using Bokeh and Javascript. Our atlas resource allows users to visualize the expression pattern of individual cells by providing a gene ID of interest and using the built-in graphical interface toolset. A factor plot displays expression levels within the parasite population based on cell cycle status, cluster identities, days post induction, or other categories. A sub-panel on the bottom left shows additional information for each input gene including translated product, mean, and standard deviation of the expression. Users can selectively highlight a subset of parasites by using a simple click-and-drag interface for further analysis. Example use cases are provided in *Figure 6e–f*. We have also made the raw data available through the atlas website. We hope this will help make our data and results readily accessible for others interested in exploring *Toxoplasma* parasitology.

## Discussion

We describe here single-cell RNA sequencing (scRNA-seq) for measurement of mRNA transcripts from individual extracellular in vitro Toxoplasma gondii, an obligate intracellular protozoan parasite. The results show that scRNA-seq can reveal intrinsic biological variation within an asynchronous population of parasites. Two types of biological variation could be seen in our asynchronous populations: cell cycle progression and asexual differentiation. We found the existence of two distinct 1N transcriptional states in cycling parasites which we call $G_1a$ and $G_1b$, concurring with what was previously reported in bulk analyses of *Toxoplasma* (*Behnke et al., 2010*). Interestingly, bradyzoites are found predominantly in $G_1b$ but not in $G_1a$, suggesting the possibility of a putative checkpoint between these two phases that may also play a role in regulating the developmental transition. Our data further shows a small fraction of bradyzoites to be cycling which supports the hypothesis that bradyzoites can in fact divide (*Sinai et al., 2016*). Our results showed a very strong correlation between cell cycle and expression of genes encoding proteins in various subcellular organelles, as noted previously using synchronized bulk populations (*Behnke et al., 2010*). The results here, however, show an even more dramatic and extreme dependence on cell cycle, allowing refinement of approaches that use such timing to predict a given protein's ultimate organellar destination in the cell (*Camejo et al., 2014*). They also extend such analyses to the Type II strains, Pru and ME49, which have not previously been examined in this way.

In addition to the above, we observed some striking and unexpected heterogeneity within asexually developing parasites. We discovered a cluster of cells, labeled P6, in the differentiating Pru parasites that is distinct from the rest of the alkaline-induced population of cells. Constituting 21% of the alkaline-induced population, the P6 cluster is marked by a set of genes that were previously detected by bulk transcriptomics in bradyzoites of tissue cysts (*Ramakrishnan et al., 2019*). Remarkably, while most of these genes have unknown functions, we identified an enriched gene with a predicted AP2 domain, which may contribute to the unique expression pattern observed in this group of parasites. We found that the P6 expression profile is intermediate to P2 tachyzoites and P1 bradyzoite clusters. Interestingly, the genes enriched in P6 overlap with a subset of canonical bradyzoite marker genes including *LDH2* and *SRS35A*, albeit expressed at a lower level than in P1 (*Figure 3c*). In addition, we observed a gradual increase in the proportion of P6 cells as induction proceeded from day 3 to day 7. Taken together, one possible explanation for the emergence of P6 cluster is a reverted conversion from bradyzoites to tachyzoites in which alkaline stress fails to maintain the bradyzoite state. Our data and previous reports are consistent with this interpretation (*Weiss et al., 1998*). On the other hand, we cannot rule out the possibility that this cluster is developmentally 'confused' by the presence of a general stressor such as alkaline. RNA velocity analysis in the Pru data does not reveal a strong transcriptional flow between P1 and P6. Rather, P6 appears to transcriptionally transition from P2 tachyzoites. Thus, the P1 bradyzoites and P6 parasites are either distinct and separate developmental trajectories, or the transition from P1 to P6 is a rapid and rare event. Regardless, our results reflect a surprising diversity in an asexually transitioning population of

*Toxoplasma*. Future measurement of single parasites isolated from in vivo sources coupled with genetic manipulation of the parasite genome, will further clarify the underlying developmental states that we identified here.

To quantify the variation of *SRSs*, which are generally expressed at low copy number, we performed 96-well Smart-seq2, which greatly improved measurement sensitivity over the 384-well format, likely due to changes in the input mRNA concentration. For scRNA-seq of pathogens, which tend to have smaller size and lower mRNA content than mammalian cells, we think a careful selection of the measurement approach is necessary based on consideration of throughput and measurement sensitivity, between which there is often a tradeoff. Combined with a novel approach that we developed based on random permutation and K-nearest neighbor (KNN) averaging, we were able to quantify the association of gene expression variation to known biological processes, like cell cycle and development. We discovered that *Toxoplasma* exhibits unexplained, sporadic variation in the expression of most *SRSs*, which may have biological implications. For example, it could expand the mode of interactions with the host and be the result of strong selective pressure to maximize invasion efficiency and transmission in a variety of different host species of cell types. Maintaining a large phenotypic diversity can be beneficial in ensuring at least some members will be able to invade the cells it encounters and/or evade adaptive immune response, enabling propagation in whatever the host environment encountered.

Very surprisingly, our scRNA-seq analysis identified an atypical co-expression pattern in an in vitro RH 'tachyzoite' that is indicative of sexual development, which has not been previously observed in these culture conditions. This suggests that at least the beginnings of sexual developmental can spontaneously occur even in the absence of the cat intestinal environment or other chemical cues. Combined with transient expression experiments, the data from this cell enabled us to show that AP2IX-1 is sufficient to drive a switching of surface antigen expression toward that resembling the sexual stages of the parasite. Assuming this change in mRNA abundance translates into a change in protein levels of TGGT1_222305, which is predicted to contain a transmembrane domain, and SRS22C, which is strongly indicated to be a surface antigen like the rest of SRS, our data indicate that AP2IX-1 contributes to remodeling of the surface antigen repertoire during differentiation. Furthermore, this suggests AP2IX-1 may play a causal role in controlling the sexual differentiation of *Toxoplasma*. Considering that the family of AP2 transcription factors was originally found to regulate stress response and floral sexual differentiation in plants, our finding suggests the biological role of AP2 family in *Toxoplasma* is evolutionarily conserved. Switching of surface antigens in parasites may be particularly favorable to parasites under stressful conditions, perhaps including the stress of an immune response, thereby enabling evasion of host immunity. Regardless, these results show that scRNA-seq can reveal rare parasite variants that are, presumably, a result of spontaneous epigenetic changes similar to what has been described in cancer cells (*Litzenburger et al., 2017*) or transcriptional noise which was previously characterized in *Escherichia coli* (*Elowitz et al., 2002*). An important difference from the situation with cancer cells, however, is that these individual variants may be non-viable and so impossible to obtain as a stable line; thus scRNA-seq may be uniquely able to provide a detailed understanding of their very interesting and informative gene expression. In this case, a previously uncharacterized AP2 factor was revealed and shown to be responsible for regulating at least some of the genes that were uniquely expressed in this variant, relative to the remainder of the tachyzoites in this population. Expanding the number of AP2 transcription factors (totaling over 68 of them) analyzed in this way could enable the deduction of the sets of 'regulons' in this parasite. Thus, even though these rare variants may be non-viable 'biological freaks', they can be highly informative and would be completely undetectable in bulk measurement.

Recently, cross-species analysis of scRNA-seq datasets has attracted considerable interest (*Butler et al., 2018*; *Ding et al., 2019*). Our study provides the first comparative analysis of developmental processes between two apicomplexans, both of which cause prevalent and potentially devastating diseases. While *Plasmodium* and *Toxoplasma* undergo distinct modes of cell replication and asexual development, we identified cross-species clusters that share significant similarity in the expression of orthologous genes. We discovered that the timing of expression in gene sets involved in cell cycle are conserved in the erythrocytic cycle of *Plasmodium berghei*. Lastly, we have made the datasets of our study available by creating an interactive and easily accessible web-browser. We hope this encourages other individuals interested in single-cell parasitology to actively explore our dataset without having expertise in programming or bioinformatics. Building on the work described

here, which lays a foundation for a detailed understanding of the parasite itself, we anticipate future, single-cell co-transcriptomic sequencing of both the host cell and the parasite as a potentially powerful approach to further deconstruct the complexity of parasite-host interactions.

# Materials and methods

## Key resources table

| Reagent type (species) or resource | Designation | Source or reference | Identifiers | Additional information |
|---|---|---|---|---|
| Cell line (Toxoplasma gondii) | ME49 | PMID:15664907 | | |
| Cell line (Toxoplasma gondii) | Pru | PMID:18347037 | | |
| Cell line (Toxoplasma gondii) | RH mCherry/RH | This work. | | |
| Cell line (Toxoplasma gondii) | RH Δhxgprt | PMID:8662859 | | |
| Cell line (Toxoplasma gondii) | RH GFP | Gift of Michael W Panas. | | |
| Recombinant DNA reagent | pGRA-AP2I × 1-V5 | This work. | | Constitutive expression plasmid carrying AP2I × 1 in tandem fusion to V5 tag. |
| Chemical compound, drug | Propidium iodide (PI) | ThermoFisher | P3566 | |
| Chemical compound, drug | Sytox Green | ThermoFisher | S7020 | |
| Chemical compound, drug | Fixable blue dead cell stain kit | ThermoFisher | L34962 | |
| Chemical compound, drug | Vybrant DyeCycle Violet (DCV) | ThermoFisher | V35003 | |
| Chemical compound, drug | VECTASHIELD Antifade Mounting Medium with DAPI | Vector Laboratories | H-1200–10 | |
| Chemical compound, drug | Nuclease-free water | IDT | 11-04-02-01 | For SmartSeq2 protocol |
| Chemical compound, drug | recombinant RNase inhibitor | Takara Clonetech | 2313A | For SmartSeq2 protocol |
| Chemical compound, drug | 10 mM dNTP | ThermoFisher | R0194 | For SmartSeq2 protocol |
| Chemical compound, drug | ERCC RNA Spike-in Mix | ThermoFisher | 4456740 | For SmartSeq2 protocol |
| Chemical compound, drug | 10% Triton X-100 | Sigma-Aldrich | 93443 | For SmartSeq2 protocol |
| Chemical compound, drug | Buffer EB (elution buffer) | QIAGEN | 19086 | For SmartSeq2 protocol |
| Chemical compound, drug | AMPure XP nucleic acid purification beads | Beckman Coulter | A63880 | For SmartSeq2 protocol |
| Chemical compound, drug | Nucleofector Solution | Lonza | P3 Primary Cell solution | For transient transfection of Toxoplasma gondii |
| Antibody | Rabbit anti-SAG1 polyclonal antibody | PMID:15944311 | | (1:500) |
| Antibody | Mouse anti-V5 tag monoclonal antibody | Invitrogen | R960-25 | (1:1000) |

*Continued on next page*

*Continued*

| Reagent type (species) or resource | Designation | Source or reference | Identifiers | Additional information |
|---|---|---|---|---|
| Antibody | Goat polyclonal Alexa 488 Fluor-conjugated secondary antibodies | Invitrogen | A28175 | (1:1000) |
| Commercial assay, kit | SsoAdvanced Universal SYBR Green Supermix | Bio-rad | 1725271 | qPCR mastermix |
| Sequence-based reagent | All oligos used in this study | See 'supplementary_file1_oligos.csv' | | |
| Software | Analysis algorithm | www.github.com/xuesoso/singleToxoplasmaSeq | | |
| Software | Interactive browser | st-atlas.org | | |

## Cell and parasite culture

All *Toxoplasma gondii* strains were maintained by serial passage in human foreskin fibroblasts (HFFs) cultured at 37 C in 5% $CO_2$ in complete Dulbeco's Modified Eagle Medium (cDMEM) supplemented with 10% heat-inactivated fetal bovine serum (FBS), 2 mM L-glutamine, 100 U/ml penicillin, and 100 ug/ml streptomycin. *T. gondii* strains used in this study were RH, Pru-GFP, and ME49-GFP-luc.

## In vitro bradyzoite switch protocol

Differentiation to bradyzoite was induced by growth under low-serum, alkaline conditions in ambient (low) $CO_2$ as previously described (*Weiss et al., 1995*). Briefly, confluent monolayers of HFFs were infected with tachyzoites at a multiplicity of infection (MOI) of 0.025 in RPMI 1640 medium (Invitrogen) lacking sodium bicarbonate and with 1% FBS, 10 mg/ml HEPES, 100 U/ml penicillin, and 100 g/ml streptomycin at pH 8.2. The infected HFFs were cultured at 37°C without supplemented $CO_2$.

## Preparation of parasites for Fluorescence Activated Cell Sorting (FACS)

HFF monolayers infected with parasites overnight were scraped, and the detached host cells were lysed by passing them through a 25-gauge needle three times or a 27-gauge needle six times. The released parasites were spun down at 800 rpm for 5 min to pellet out host cell debris, and the supernatant was spun down at 1500 rpm for 5 min to pellet the parasites. The parasites were then resuspended in 500 μL of FACS buffer (1x phosphate-buffered saline, PBS, supplemented with 2% FBS, 50 ug/ml DNAse I, and 5 mM $MgCl_2$*$6H_2O$), passed through both a 5 μm filter and a filter cap into FACS tubes, and stored on wet ice until it was time to sort. In samples stained for DNA content, the parasites were resuspended in 500 μL of FACS buffer plus 1.5 μL of Vybrant DyeCycle Violet (from ThermoFisher, catalog number V35003) and incubated at 37 C and 5% $CO_2$ for 30 min.

The parasites were also stained with either propidium iodide (PI), Sytox Green, or the live/dead fixable blue dead cell stain kit (catalog number L34962) prior to sorting in order to distinguish live cells from dead cells. To stain with PI, 10 μL of 0.5 mg/ml PI was added to every 500 μL of parasite suspension in FACS buffer, and the parasites were incubated covered on ice for at least 15 min. To stain with Sytox Green, 1 drop of Sytox Green per ml was added to the parasite suspension in FACS buffer, and the parasites were incubated at room temperature for at least 15 min. To stain with the live/dead fixable blue dead cell stain kit, 1.5 μL of the kit's viability dye was added to every 500 μL of parasites along with the secondary antibody, and parasites were washed and resuspended in FACS buffer as usual.

## FACS of parasites

Eight mL of lysis buffer was prepared by mixing together: 5.888 mL of water, 160 μL recombinant RNase inhibitor (Takara Clonetech), 1.6 mL of 10 mM dNTP (ThermoFisher), 160 μL of 100 uM oligo-dT (iDT; see attached *Supplementary file 1* for oligos), 1:600,000 diluted ERCC spike-in (ThermoFisher), and 32 μL of 10% Triton X-100. All reagents are declared RNase free. Lysis plates were prepared by dispensing 0.4 μL of lysis buffer into each well of a 384 well hard-shell low profile PCR plate (Bio-rad) using liquid handler Mantis (Formulatrix). Single parasites were sorted using the

Stanford FACS Facility's SONY SH800s sorter or BD Influx Special Order sorter into the 384-well plates loaded with lysis buffer. Single color and colorless controls were used for compensation and adjustment of channel voltages. The data were collected with FACSDiva software and analyzed with FlowJo software. RH parasites were index sorted with fluorescence signal of cell permeable DNA stain, DyeCycle Violet.

## Single-Toxoplasma cDNA synthesis, library preparation, and sequencing

Smart-seq2 protocol was carried out as previously described (Picelli et al., 2014) using liquid handlers Mantis and Mosquito (TTP Labtech) with slight modifications. 384-well Smart-seq2 was performed with a 2 µL final reaction volume, while 96-well format was carried out in 25 µL final reaction volume as recommended by the original protocol. For 384-well Smart-seq2, we performed 19 rounds of cDNA pre-amplification after reverse transcription with oligo-dT primers. Each well is then diluted with 1 to 4 v:v in RNAse free elution buffer (QIAgen) to a total volume of 8 µL. For 96-well Smart-seq2, we performed 30 rounds of cDNA pre-amplification after reverse transcription. PCR is performed with 'IS_PCR primers'. Each well is then purified with Ampure XP beads at 0.8X volume ratio and resuspended in 20 µL RNAse free elution buffer. We measured the size distribution and concentration of each well in 96-well plate using Fragment Analyzer High-sensitivity NGS kit (Agilent). We normalized the concentration of cDNA of each well to a concentration of 0.4 ng/µL. Finally, for both 384-well and 96-well Smart-seq2 measurements, we conducted library preparation with in-house Tn5 tagmentation using custom cell barcode and submitted for 2 × 150 bp paired-end sequencing on NovaSeq 6000 at the Chan Zuckerberg Biohub Genomics core. All primer sequences are provided as a supplementary file.

## AP2IX-1 transient expression

The pGRA-AP2I × 1-V5 plasmid for AP2I × 1 transient expression was created using Gibson assembly (NEB) from the pGRA-V5 (Panas et al., 2019). RH parasites were transfected with pGRA-AP2I × 1-V5 using the Amaxa 4D Nucleofector (Lonza). Tachyzoites were mechanically released in PBS, pelleted, and resuspended in 20 µL P3 Primary Cell Nucleofector Solution (Lonza) with 7 or 15 µg DNA for transfection. After transfection, parasites were allowed to infect HFFs in DMEM. After 18–20 hr of infection, parasites were prepared for Immunofluorescence Assay (IFA) or qRT-PCR. To compute statistical independence between the transfected and control samples, we applied Student's t-test by assuming unequal variance. One sigma (*) indicates one standard deviation in mean difference assuming null hypothesis.

## Immunofluorescence assay (IFA) quantification

Monolayers of infected cell on glass coverslips were fixed with cold methanol for 12 min. Samples were washed with PBS and blocked using 3% bovine serum albumin (BSA) in PBS for at least 30 min. SAG1 was detected with rabbit anti-SAG1 polyclonal antibody and V5 was detected with mouse anti-V5 tag monoclonal antibody (Invitrogen). Primary antibodies were detected with goat polyclonal Alexa Fluor-conjugated secondary antibodies (Invitrogen). Primary and secondary antibodies were both diluted in 3% BSA in PBS. Coverslips were incubated with primary antibodies for 30 min, washed, and incubated with secondary antibodies for 30 min. Vectashield with DAPI stain (Vector Laboratories) was used to mount the coverslips on slides. Fluorescence was detected using wide-field epifluorescence microscopy and images were analyzed using ImageJ. All images shown for any given condition/staining in any given comparison/dataset were obtained using identical parameters.

## Quantitative polymerase chain reaction (qPCR)

To quantify the purity of single parasite sort and to ensure the cDNA synthesis reaction was not saturated, GFP, mCherry, or SAG1 mRNA expression were measured using commercial qPCR mastermix, SsoAdvanced Universal SYBR Green mastermix (Bio-rad). Briefly, 0.1 µL of diluted cDNA was added in a total of 2.1 µL reaction volume per well on a 384 well plate with qPCR mastermix and 200 nM PCR primers. The reaction was incubated on a Bio-rad qPCR thermal cycler with the following programs: 5 min of 95℃, 45 cycles of 95℃ for 5 s and 56℃ for 1 min, and imaging. To quantify the transcriptional effects of AP2IX-1 transient expression in RH parasites, infected cell monolayers were first lysed with Trizol (Invitrogen) and RNA was extracted using standard molecular biology technique.

cDNA from each sample was generated with Smart-seq2 protocol using 20 µL total reaction volume and roughly 200 ng RNA input followed by 0.8X AMPure XP beads (Beckman Coulter) purification. For qPCR, 1 ng of cDNA was added in a total of 2.1 µL reaction volume per well on a 384 well plate with qPCR mastermix and 200 nM PCR primers as described above. The reaction was incubated on a Bio-rad qPCR thermal cycler with the following programs: 5 min of 95°C, 60 cycles of 95°C for 5 s and 61°C for 30 s, and imaging. Each gene was measured four times with samples collected from at least two separate wells. The transfection and qRT-PCR experiments were performed twice in separate experiments. Fold change of gene expression was calculated as shown previously (*Livak and Schmittgen, 2001*) using ACT1 expression as an internal control for samples. All primer sequences are provided in *Supplementary file 1*.

## Sequencing alignment

BCL output files from sequencing were converted into gzip compressed FastQs via a modified bcl2fastq demultiplexer which is designed to handle the higher throughput per sequencing run. To generate genome references with spike-in sequences, we concatenated T. gondii ME49 NCBI genome assembly version 11/1/2013 or T. gondii GT1 T NCBI Genome assembly version 7/19/2013 genome references with ERCC sequences (obtained from ThermoFisher website). The raw fastq files from sequencing are aligned to the concatenated genomes with STAR aligner (version 2.6.0c) using the following settings: '−readFilesCommand zcat −outFilterType BySJout −outFilterMutli-mapNmax 20 −alignSJoverhangMin 8 −alignSJDBoverhangMin 1 −outFilterMismatchNmax 999 −outFilterMismatchNoverLmax 0.04 −alignIntronMin 20 −alignIntronMax 1000000 −alignMatesGapMax 1000000 −outSAMstrandField intronMotif −outSAMtype BAM Unsorted −outSAMattributes NH HI AS NM MD −outFilterMatchNminOverLread 0.4 −outFilterScoreMinOverLread 0.4 −clip3pAdapterSeq CTGTCTCTTATACACATCT −outReadsUnmapped Fastx'. Transcripts were counted with a custom htseq-count script (version 0.10.0, https://github.com/simon-anders/htseq) using ME49 or RH GFF3 annotations (version 36 on ToxoDB) concatenated with ERCC annotation. Instead of discarding reads that mapped to multiple locations, we modified htseq-count to add transcript counts divided by the number of genomic locations with equal alignment score, thus rescuing measurement of duplicated genes in the *Toxoplasma* genome. Parallel jobs of STAR alignment and htseq-count were requested automatically by Bag of Stars (https://github.com/iosonofabio/bag_of_stars) and computed on Stanford high-performance computing cluster Sherlock 2.0. Estimation of reads containing exonic and intronic regions is computed with Velocyto estimation on the BAM output files and requested automatically by Bag of Velocyto (https://github.com/xuesoso/bag_of_velocyto) on Sherlock 2.0. Gene count matrix is obtained by summing up transcripts into genes using a custom python script. Scanpy velocyto package is then used to estimate transcriptional velocity on a given reduced dimension. Parameters used for generating the results are supplied as supplementary python scripts. Sample code to generate the analysis figures are provided in supplementary jupyter notebooks.

## Data preprocessing

To filter out cells with poor amplification or sequencing reaction and doublet cells, we discarded cells based on gene counts (>0 reads), total reads sum, percent reads mapped to *Toxoplasma* genome, percent ERCC reads, and percent ribosomal RNA reads. We reported '% mapped' based on the meta-alignment output from STAR aligner. We checked for some of the unmapped reads on BLASTn and found the majority of them to map to Toxoplasma 28S ribosomal RNA. Next, we filtered 'ribosomal RNA' genes from the gene count matrix. Gene count matrices are normalized as counts per median (CPM):

$$X_{norm} = \frac{x}{\sum(x)} \cdot medium\left(\sum(x)\right) \tag{1}$$

where $X$ is the gene count matrix, *sum(X)* is the read sum for each cell, and *median(sum(X))* is the median of read sums. Normalized data are added with a pseudocount of 1 and log transformed (e.g. $\log_2(X_{norm}+1)$). To determine the detection limit (e.g. 50% detection rate), we modeled the detection probability of ERCC standards with a logistic regression as a function of spike-in amount (*Svensson et al., 2017*).

We calculated an estimate of absolute molecular abundance for all genes by fitting a linear regression to ERCC spike-ins:

$$log_2(y) = m \cdot log_2(X_{norm} + 1) + b \tag{2}$$

where $X_{norm,\ ERCC>0.5}$ is the observed CPM value for ERCC spike-ins above the detection limit, $Y$ is the amount of ERCC spike-in, $m$ is the regression coefficient, and $b$ is the intercept. To reduce the influence of measurement noise, we fit the model only to ERCC spike-ins with mean expression above the detection limit.

## Cell cycle analysis and annotation

To determine the transcriptional variation associated with cell cycle, we applied Self-Assembling Manifolds (SAM) (*Tarashansky et al., 2019*) to filter for highly dispersed gene sets (>0.35 SAM weights) in asynchronous RH population. Principal components analysis (PCA) is then applied to the filtered and normalized RH data, and the nearest neighbor graph (K = 50) is computed using 'correlation' as a similarity metric. We identified the putative 'G1' clusters with 1N based on DNA content stain. Parasites in 'G1' cluster are further sub-clustered with Louvain Clustering, in which we identified 'G$_1$a' and 'G$_1$b' clusters with distinct transcriptional profiles. Pearson correlation between single-cell and bulk transcriptomic data is computed between bulk assignment (*Behnke et al., 2010*) and the scRNA-seq cluster assignment through which each cluster is uniquely assigned with a cell cycle state. To quantify genes that are differentially expressed across cell cycle clusters, we applied Kruskal-Wallis test. Genes are considered differentially expressed if their p-values are less than 0.05 and they are at least 2-fold over-expressed in a cluster compared to the average expression level of other clusters. We computed differential expression across all cell cycle clusters as well as between the 'G$_1$a' and 'G$_1$b' clusters; the results are uploaded as *Supplementary files 2* and *3*, respectively. To enable cell cycle assignment transfer from RH to Pru and ME49 data, we implemented a random forest classification model trained on RH data. Briefly, this is done by training a model with 1000 estimators on L2-normalized RH expression data containing only cell cycle associated genes in a 60–40 split scheme. Then the model is applied to predict cell cycle labels of L2-normalized Pru or ME49 data containing the homologous cell cycle associated genes. The testing accuracy was over 95%.

## Pseudotime construction and clustering

Pseudotime analysis is conducted with Monocle two package in R on preprocessed dataset with highly dispersive genes as described previously. A cell in 'G$_1$a' is designated as the root cell, and all other cells are placed after this cell in order of their inferred pseudotime. To cluster genes based on their pseudotime expression pattern, high frequency patterns are removed through a double spline smoothing operation. The interpolated expression matrix is then normalized by maximum expression along pseudotime such that the maximum value of gene expression along pseudotime is bound by 1. We then applied agglomerative clustering on this interpolated and normalized expression matrix using 'correlation affinity' as similarity metric and 'average linkage' method to predict three distinct clusters of genes.

## Projection dependence scoring

To quantify the dependence of expression variation on a two-dimensional projection, we developed a novel approach based on k-nearest neighbor (KNN) averaging. First, a KNN graph is computed by locating nearest neighborhood in a projection using euclidean distance. We then generated a null expression matrix by shuffling the gene expression matrix along each cell column, such that its correlation with respect to the coordinate on projection is completely lost. Next, we compute an updated gene expression value by taking the average of expression values across the KNN. This is equivalent to:

$$X_{KNN} = \frac{M}{k} \cdot X_{norm} \tag{3}$$

where $X_{KNN}$ is the updated KNN averaged expression, $M$ is the nearest-neighbor graph with $k$ being the number of nearest neighbor, and $X$ is the log-transformed CPM of observed or null expression matrices. We chose a $k$ of 5 for all our analysis as varying $k$ did not have a large effect on the results

(data not shown). In our experiments, we have shown that the first two principal components (PCs) of PCA on RH correspond to the projection of cell cycle progression, and a two-dimensional UMAP projection of Pru corresponds to asexual development and cell cycle progression. We thus computed $X_{KNN}$ for both the original, observed expression matrix and the shuffled, null matrix on either projection to reflect dependence on cell cycle progression and/or asexual development. $X_{KNN}$ is further normalized to have identical sum as the original expression values. A Kolmogorov-Smirnoff two sample test is then computed between the normalized $X_{KNN}$ of the observed matrix and that of the shuffled matrix based on 100 random permutations. The projection-dependence score for each gene is then computed as:

$$S_g = \sqrt{-log(\bar{p}_g)} \tag{4}$$

where $S_g$ is the projection-dependence score for gene $g$ and $\bar{p}_g$ is the average p-values of 100 tests. We present $S_g$ normalized by the maximum score within each respective data set.

## Comparative analysis of plasmodium and Toxoplasma scRNA-seq

To integrate scRNA-seq data of *Plasmodium berghei* from Malaria Atlas (*Howick et al., 2019*) with our *Toxoplasma* Pru dataset (measured both induced and induced population in 384-well), we first identified one-to-one orthologous genes obtained from PlasmoDB (https://plasmodb.org/) and ToxoDB (https://toxodb.org/toxo/). Next, we filtered each dataset with the ortholog genes. Using scanpy library, we filtered for the intersect of genes that are within the top 800 most dispersed genes in each dataset, resulting in a list of 403 genes. Finally, we used Scanorama (*Hie et al., 2019*) with default parameter settings to integrate the two datasets. We calculated co-clustering similarity as follows. We first computed a Leiden (*Traag et al., 2019*) clustering on the integrated graph and returned a cluster co-occurrence matrix to the original cluster assignment in *Plasmodium berghei* ('ShortenedLifeStage4') or *Toxoplasma* Pru ('cell_cylcle'). Then, the dot product between the two matrices was calculated and normalized such that it has a maximum of one.

## Acknowledgements

We thank Fabio Zanini, Felix Horns, and Geoff Stanley for illuminating discussion and advice to YX on experiments and analysis. We thank Saroja Korullu, Robert Jones, and Vickie Lin for assistance with library preparation and sample submission. We thank Meredith Weglarz and Lisa Nichols at the Stanford Beckman FACS facility for assistance with FACS. We thank Michael W Panas for providing Toxoplasma GFP strain used in this study. This study is supported by National Institute of Health (NIH) RO1 AI021423, AI129529, and Chan Zuckerberg Biohub. YX and TCT are supported by Stanford Interdisciplinary Graduate Bio-X Fellowships. SR is supported by NIH F30 AI124589-03. AF is supported by NIH 5T32AI007328-30 and a Gilliam Fellowship for Advanced Study from Howard Hughes Medical Institute.

## Additional information

### Funding

| Funder | Grant reference number | Author |
|---|---|---|
| Stanford University | Stanford Interdisciplinary Graduate Bio-X Fellowships | Yuan Xue Terence C Theisen |
| National Institutes of Health | F30 AI124589-03 | Suchita Rastogi |
| National Institutes of Health | 5T32AI007328-30 | Abel Ferrel |
| Howard Hughes Medical Institute | Gilliams Fellowship for Advanced Study | Abel Ferrel |
| National Institutes of Health | RO1 AI21423 | John C Boothroyd |
| National Institutes of Health | RO1 AI29529 | John C Boothroyd |
| Chan Zuckerberg Biohub | | Stephen R Quake |

The funders had no role in study design, data collection and interpretation, or the decision to submit the work for publication.

## Author contributions
Yuan Xue, Conceptualization, Data curation, Software, Formal analysis, Validation, Investigation, Visualization, Methodology, Writing - original draft, Project administration, Writing - review and editing; Terence C Theisen, Resources, Formal analysis, Validation, Investigation, Methodology, Writing - review and editing; Suchita Rastogi, Resources, Investigation, Methodology, Writing - review and editing; Abel Ferrel, Resources, Methodology, Writing - review and editing; Stephen R Quake, Resources, Supervision, Funding acquisition, Writing - review and editing; John C Boothroyd, Conceptualization, Resources, Supervision, Funding acquisition, Writing - review and editing

## Author ORCIDs
Yuan Xue (iD) https://orcid.org/0000-0002-7846-4273
Stephen R Quake (iD) https://orcid.org/0000-0002-1613-0809
John C Boothroyd (iD) https://orcid.org/0000-0001-9719-745X

## Decision letter and Author response
Decision letter https://doi.org/10.7554/eLife.54129.sa1
Author response https://doi.org/10.7554/eLife.54129.sa2

# Additional files

## Supplementary files

• Supplementary file 1. A list of sequence-based reagents (e.g. oligos) used in this study.

• Supplementary file 2. A table of summary statistics and fold change of genes that are found to be differentially expressed between 'G1 a' and 'G1 b' parasites in RH. Genes are considered differentially expressed only if the adjusted p-values of Kruskal-Wallis test are less than 0.05 and that they are at least 2-fold over-expressed in either of the cell states.

• Supplementary file 3. A table of summary statistics and fold change of genes that are found to be differentially expressed between across all cell cycle states in RH parasites. Genes are considered differentially expressed only if the adjusted p-values of Kruskal-Wallis test are less than 0.05 and that they are at least 2-fold over-expressed in any of the cell states.

• Supplementary file 4. A table summarizing the hierarchical clustering results and encoded gene products. Clustering is performed on smoothened gene expression based on biological pseudotime in RH parasites.

• Supplementary file 5. A table of summary statistics and fold change of genes that are found to be differentially expressed between across all clusters in Pru parasites. Genes are considered differentially expressed only if the adjusted p-values of Kruskal-Wallis test are less than 0.05.

• Supplementary file 6. A table of summary statistics and fold change of genes that are found to be differentially expressed between across all clusters in ME49 parasites. Genes are considered differentially expressed only if the adjusted p-values of Kruskal-Wallis test are less than 0.05.

• Transparent reporting form

## Data availability

Analysis scripts, preprocessing scripts, and instructions to obtain the processed data are provided on https://github.com/xuesoso/singleToxoplasmaSeq (copy archived at https://github.com/elifes-ciences-publications/singleToxoplasmaSeq). A sample jupyter notebook that regenerates some of the analysis and figures is provided in the Git repository. Raw fastq files and processed data are deposited on SRA and GEO repository (GEO number: GSE145080).

The following datasets were generated:

**Database and**

| Author(s) | Year | Dataset title | Dataset URL | Identifier |
|---|---|---|---|---|
| Xue Y, Theisen T, Rastogi S, Ferrel A, Quake SR, Boothroyd JC | 2020 | A single-parasite transcriptional atlas of asexual development in Toxoplasma gondii reveals novel control of antigen expression | https://www.ncbi.nlm.nih.gov/geo/query/acc.cgi?acc=GSE145080 | NCBI Gene Expression Omnibus, GSE145080 |
| Xue Y, Theisen T, Rastogi S, Ferrel A, Quake SR, Boothroyd JC | 2020 | Data from: A single-parasite transcriptional landscape of Toxoplasma gondii reveals novel control of antigen expression | http://dx.doi.org/10.5061/dryad.kprr4xh17 | Dryad Digital Repository, 10.5061/dryad.kprr4xh17 |

The following previously published datasets were used:

| Author(s) | Year | Dataset title | Dataset URL | Database and Identifier |
|---|---|---|---|---|
| Behnke MS, Wootton JC, Lehmann MM, Radke JB, Lucas O, Nawas J, Sibley LD, White MW | 2010 | Coordinated progression through two subtranscriptomes underlies the tachyzoite cycle of toxoplasma gondii | https://www.ncbi.nlm.nih.gov/geo/query/acc.cgi?acc=GSE19092 | NCBI Gene Expression Omnibus, GSE19092 |
| Ramakrishnan C, Maier S, Walker RA, Rehrauer H, Smith NC, Grigg ME, Deplazes P, Helh AB | 2018 | Transcriptomics of Toxoplasma gondii enteroepithelial stages | https://www.ncbi.nlm.nih.gov/geo/query/acc.cgi?acc=GSE108740 | NCBI Gene Expression Omnibus, GSE108740 |

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
