## [Decision Letter]

**Acceptance summary:**

This a welcome and timely study of individual parasite gene expression the asexual phase of several stains of *Toxoplasma gondii* covering both tachyzoite and bradyzoite development with a SmartSeq2 approach, which represents a significant contribution to the field. It leverages and confirms earlier cell cycle work of others while also nicely informing on biological differences between individual parasites and strains during asexual growth and differentiation. The work also illustrates how new regulatory pathways can be identified and the extent to which they can be conserved over vast evolutionary timescales.

**Decision letter after peer review:**

Thank you for submitting your article "A single-parasite transcriptional landscape of *Toxoplasma gondii* reveals novel control of antigen expression" for consideration by *eLife*. Your article has been reviewed by three peer reviewers, and the evaluation has been overseen by Dominique Soldati-Favre as the Senior and Reviewing Editor. The following individuals involved in review of your submission have agreed to reveal their identity: Jessica Kissinger (Reviewer #2).

The reviewers have discussed the reviews with one another and the Reviewing Editor has drafted this decision to help you prepare a revised submission.

Summary:

The authors have performed rigorous controls and experimental design and made the data freely available in both its raw form and through a user-friendly interface. The discoveries are significant and it is tantalizing to consider what other insights may be gained with additional and deeper exploration of strains with phenotypic differences. Overall this is an outstanding study and an important resource for the community that deserves publication once suggestions/reservations about some aspects of the analysis and the manuscript will have be addressed.

Essential revisions:

1) Several points of clarifications

– As not all individual single-cell experiments were successful, it may make more sense, except when discussing the success rate, to utilize the numbers of parasite datasets (as opposed to parasites) that could actually be compared, e.g. paragraph five of subsection “Hidden heterogeneity in asexually developing Toxoplasma”, since only 1552 datasets could be compared.

– RH genome/gff files that were used represent only 4 Mb of sequence data including and the gff file only contains annotation for the plastid genome sequence. Paragraph two of subsection “Technical validation of single-parasite sorting and sequencing” states that RH reads were mapped to the GT1 genome sequence but this is not reflected in the Materials and methods where only TgME49 and RH are mentioned as reference sequences. The manuscript should more clearly represent exactly which genome sequences and sources were utilized in the methods. See also subsection “Sequencing alignment”.

– It is great that the authors consider the "multiple-mapping problem" and devise a work-around. There is however another issue related to genome misassembly and compressed multi-gene families or recent segmental duplications. This is an issue for most genome sequences and cannot be resolved here but it would be good to acknowledge the effect that missing gene family members may have on the analysis of the results. Specifically, did the genome sequences used contain the "unassembled contigs".

– Subsection “An open-source interactive resource for visualizing single-Toxoplasma atlas” – any thoughts about the longer-term sustainability of the atlas resource? Have the sequence data been deposited in the SRA read archive?

– Discussion paragraph three – why is mRNA concentration affected by the size of the well and reaction volume used? there is still only a single cell in the assay, but the reaction volumes are greater. Saturation was proven to not be a problem with the smaller 384-well format but here, sensitivity to low copy number is favored. Please clarify.

– Figure 1—figure supplement 2, what is panel b really telling us? are differences in genome assembly or annotation skewing the results? Also by ORF do you really mean CDS? how were these obtained? they are not mentioned in the Materials and methods.

2) The authors have chosen some surprising parameters in the mapping:

– The star aligner parameters for max intron and mate gap size is set to 1Mbp, this has been found to lead to some incorrect mapping in other systems and can result in low level misattributed reads; whilst this is not likely to sway the presented analysis in a significant way, it should be checked.

– The choice to include and distribute multiply mapped reads of equivalent quality across different genes is somewhat problematic as it will result in one initial read to be attributed to several genes which is not a true reflection of the underlying signal. The results might be particularly biased in the analysis of the multigene family. Apart from recovering a more important number of genes per cell which is not a valid aim in itself, the authors have not justified why this is needed and not demonstrated that it does not impact their downstream analyses significantly.

3) In the analysis relating the organelle-specific expression clustering, the authors successfully identify correctly and mis-attributed organellar proteins described in the literature. This approach is promising but the further clustering of pseudotime in 3 clusters seems unnecessary, hierarchical clustering of each organellar set ordered in pseudotime may be more informative. Moreover, it could be interesting to compare gene expression patterns and cluster them finely on the whole dataset so as to potentially identify proteins not yet ascribed to any organelle but who share expression patterns with those already described.

4) The bradyzoite diversity observed and the strain specific differences is a significant observation. The authors have not attempted to understand the transcriptomic circuitry that underlies decision to bifurcate to a bradyzoite fate and the strain specific differences associated with that decision. The authors hypothesize that P3 might be a state from which parasites can trifurcate into the cell cycle or either of the two separate bradyzoite clusters. This could be tested and described more granularly by further sub clustering, pseudotime ordering and branching analysis to understand the transcriptomic determinants of bifurcation into these fates.

5) The claim of antigenic switching based on a single cell with a different SRS expression pattern, although an interesting initial observation, seems over-interpreted based on the data presented. It is not clear what the author's hypothesis is with regards to this cell, i.e is it the only cell undergoing switching in the population? Why does it express a sexual stage SRS? Does the SAG1 protein signal disappear completely upon transfection with the AP2? The switching mediated by the AP-2 would need a more single cell measurement of the pattern of antigen expression (e.g. scRNA-seq of sorted parasites with different levels of the AP2), although this would be a big undertaking. The authors should either add more data to complete this observation or alternatively should critically discuss their observations and tone down their conclusions.

---

## [Author Response]

Essential revisions:1) Several points of clarifications– As not all individual single-cell experiments were successful, it may make more sense, except when discussing the success rate, to utilize the numbers of parasite datasets (as opposed to parasites) that could actually be compared, e.g. paragraph five of subsection “Hidden heterogeneity in asexually developing Toxoplasma”, since only 1552 datasets could be compared.

In Figure 1 and Results section, the number presented indeed reflects the number of parasites that we sorted and sequenced, as the reviewers surmised. But we agree with the reviewers that it will be more helpful to show the number of parasites that passed quality control and were analyzed instead. We have now made the relevant changes to figures and text in the manuscript and include both the number of sorted/sequenced parasites and analyzed parasites in the text (Results paragraph two) where appropriate to provide an estimate for the yield of single transcriptomes.

– RH genome/gff files that were used represent only 4 Mb of sequence data including and the gff file only contains annotation for the plastid genome sequence. Paragraph two of subsection “Technical validation of single-parasite sorting and sequencing” states that RH reads were mapped to the GT1 genome sequence but this is not reflected in the Materials and methods where only TgME49 and RH are mentioned as reference sequences. The manuscript should more clearly represent exactly which genome sequences and sources were utilized in the methods. See also subsection “Sequencing alignment”.

The reviewers are correct. We made a mistake in the original manuscript – the RH datasets were aligned to GT1 genome reference and gff annotations. We have provided more details regarding the sources and dates of genome references and annotations used in this study. This also addresses the issues raised by the reviewers in subsection “Sequencing alignment”.

– It is great that the authors consider the "multiple-mapping problem" and devise a work-around. There is however another issue related to genome misassembly and compressed multi-gene families or recent segmental duplications. This is an issue for most genome sequences and cannot be resolved here but it would be good to acknowledge the effect that missing gene family members may have on the analysis of the results. Specifically, did the genome sequences used contain the "unassembled contigs".

We agree that gene duplication may have an effect on our analysis results, which, as the reviewers mentioned, would be difficult for us to address in this work; however, since both the GT1 and ME49 genome references we used contain "unassembled contigs", we do not expect missing gene family members to be a major source of error on the analysis.

– Subsection “An open-source interactive resource for visualizing single-Toxoplasma atlas” – any thoughts about the longer-term sustainability of the atlas resource? Have the sequence data been deposited in the SRA read archive?

We are committed to maintaining the interactive atlas until such time as a stable, third party solution can be found for it. We will also share our resources and explorer with ToxoDB, a widely used resource for the apicomplexan and Toxoplasma community. Lastly, we have now submitted the raw fastq files and processed files for deposition on GEO and SRA repository (GEO number: GSE145080).

– Discussion paragraph three – why is mRNA concentration affected by the size of the well and reaction volume used? there is still only a single cell in the assay, but the reaction volumes are greater. Saturation was proven to not be a problem with the smaller 384-well format but here, sensitivity to low copy number is favored. Please clarify.

We are also puzzled by the difference in apparent detection sensitivity between the 96-well and 384-well plate formats. While we do not find evidence that single-parasite mRNA reaches saturation in 384-well plate format, as Figure 1—figure supplement 1B indicates, this observation is not necessarily related to detection sensitivity of the scRNA-seq method. Recent studies and our anecdotal evidence suggest that detection sensitivity of scRNA-seq methods is limited by the efficiency of the reverse transcription (RT) step. Lower efficiency of the RT reaction, as usually occurs in droplet scRNA-seq methods, leads to lower conversion of mRNA transcripts to double-stranded cDNA for downstream amplification. Hughes et al. (bioRxiv, https://doi.org/10.1101/689273, 2019) has suggested that the template switching step, which is the common second-strand cDNA synthesis mechanism in Smart-seq2 and the Seq-well methods (the method of choice in their work), is the rate limiting step of RT. In their work, inclusion of a first-strand cDNA recovery step via random hexamer priming significantly improved the detection sensitivity of low abundant transcripts. We suspect that reduction of the RT reaction volume led to changes in the reagent ratio, in particular that of template switch oligos, to the input mRNA which subsequently reduced efficiency of RT; however, further experimentation is required to definitively address the cause of this. As the original Smart-seq2 protocol was implemented for 96-well plate, it may not be a total surprise that reducing the reaction volume in 384-well plates leads to reduced performance; however, sensitivity of our 384-well Smartseq2 measurement still outperforms other droplet methods.

– Figure 1—figure supplement 2, what is panel b really telling us? are differences in genome assembly or annotation skewing the results? Also by ORF do you really mean CDS? how were these obtained? they are not mentioned in the Materials and methods.

We apologize for the confusion. Figure 1—figure supplement 2 panel B shows the number of genes detected within the analyzed subset (red colored data points in Figure 1—figure supplement 2 panel A) and the percent of reads that mapped to genes containing open reading frames (ORFs) that are predicted to be bona fide coding sequences (CDSs). We originally quantified the percentage of reads that aligned to any region of the genomic reference as “% mapped to ORF”, as this is a standard meta-output from STAR alignment software. But the reviewers are right that this is misleading and so we have updated the axis label to "% mapped" in order to better reflect the fact that aligned reads may originate from intronic, intergenic, or non-coding regions. The differences between strains in percentage mapping may be due to the differences in genome assembly, as we noticed using BLASTn that the majority of unmapped reads across different datasets aligns to 28S ribosomal RNA of ME49 genome reference. We have included additional information in Materials and Methods to clarify all these details and changed the figure to read "% mapped".

2) The authors have chosen some surprising parameters in the mapping:– The star aligner parameters for max intron and mate gap size is set to 1Mbp, this has been found to lead to some incorrect mapping in other systems and can result in low level misattributed reads; whilst this is not likely to sway the presented analysis in a significant way, it should be checked.

We agree this should be checked and so we realigned all 125 SRS genes for a random sample of 6 cells in 384-well RH samples (10099007) after removing the max intron and mate gap size parameters in STAR aligner and instead used the default settings. Our results showed that the resulting read counts were identical using our original and this new set parameters in all but 3 of the instances, i.e. of the 750 gene/cell data points, 747 were identical. The 3 instances where alignment results differed, read counts did not vary by more than 2. We are therefore confident that our analysis results are robust to the changes in the STAR alignment parameters.

– The choice to include and distribute multiply mapped reads of equivalent quality across different genes is somewhat problematic as it will result in one initial read to be attributed to several genes which is not a true reflection of the underlying signal. The results might be particularly biased in the analysis of the multigene family. Apart from recovering a more important number of genes per cell which is not a valid aim in itself, the authors have not justified why this is needed and not demonstrated that it does not impact their downstream analyses significantly.

This is an important point and one we considered and discussed extensively among ourselves while doing the analyses. The reason why, in the end, we adopted a correction scheme for multiply-mapped reads is because, as the reviewers mentioned, Toxoplasma genome is known to harbor a number of multigene families. In particular, our interest in analyzing the co-expression, or lack thereof, of SAG1-related Sequence (SRS) genes hinges on the sensitive and reliable detection of SRS genes. We thus faced a choice of increasing the false positives in gene alignment by assigning reads to all genes that could be their origin or increasing false negatives by counting only reads that were uniquely aligned to genes. As we are specifically interested in understanding the underlying reasons for why some SRS genes are expressed at "low" levels in the population, we were anxious to avoid an approach that yielded false negatives. That is, we wanted to be sure that if we did not detect the expression of a given SRS gene, it was not because it was part of a closely related gene family and as a result its transcript discarded due to ambiguous assignment. Most importantly, however, we reasoned that if such gene sequences did prove to be a major contributor for ambiguous read assignment, we would see SRS genes whose expression appeared to strongly correlate with each other. In fact, the data showed almost no co-expression of SRSs other than the super-abundant class like SAG1 and SAG2 (as shown in Figure 4), and so we believe that the false positives are few, if any. We have included additional text in the manuscript to explain this reasoning.

3) In the analysis relating the organelle-specific expression clustering, the authors successfully identify correctly and mis-attributed organellar proteins described in the literature. This approach is promising but the further clustering of pseudotime in 3 clusters seems unnecessary, hierarchical clustering of each organellar set ordered in pseudotime may be more informative. Moreover, it could be interesting to compare gene expression patterns and cluster them finely on the whole dataset so as to potentially identify proteins not yet ascribed to any organelle but who share expression patterns with those already described.

We apologize that we were not clear with the presentation of supplementary data in our original manuscript. We have provided an additional table (Supplementary file 4) that includes the pseudotime cluster assignment for all genes, which would enable readers to easily identify genes with unknown functions that may share temporal expression patterns of known gene families.

4) The bradyzoite diversity observed and the strain specific differences is a significant observation. The authors have not attempted to understand the transcriptomic circuitry that underlies decision to bifurcate to a bradyzoite fate and the strain specific differences associated with that decision. The authors hypothesize that P3 might be a state from which parasites can trifurcate into the cell cycle or either of the two separate bradyzoite clusters. This could be tested and described more granularly by further sub clustering, pseudotime ordering and branching analysis to understand the transcriptomic determinants of bifurcation into these fates.

Thank you for the suggestion. The idea of using scRNA-seq data to infer fate decision from transcriptional signature has been recently attempted in Weinreb et al. (Science, DOI: 10.1126/science.aaw3381, 2020). Combined with the use of lineage-tracing barcode, the authors concluded that intrinsic fate biases could not be determined by scRNA-seq. In the absence of clonal identity marker, we find it challenging to convince ourselves that fate determinants can be extracted from our scRNA-seq datasets without further experimental validation; however, we have made it possible for future studies to infer such determinants by providing a list of cluster-specific gene expression for Pru (Supplementary file 5) and ME49 (Supplementary file 6). Overall, we find that the RNA velocity analysis we present provides very clear indications of the flow within and between some of the subpopulations, as shown in Figure 3B.

5) The claim of antigenic switching based on a single cell with a different SRS expression pattern, although an interesting initial observation, seems over-interpreted based on the data presented. It is not clear what the author's hypothesis is with regards to this cell, i.e is it the only cell undergoing switching in the population? Why does it express a sexual stage SRS? Does the SAG1 protein signal disappear completely upon transfection with the AP2? The switching mediated by the AP-2 would need a more single cell measurement of the pattern of antigen expression (e.g. scRNA-seq of sorted parasites with different levels of the AP2), although this would be a big undertaking. The authors should either add more data to complete this observation or alternatively should critically discuss their observations and tone down their conclusions.

We are sorry for the confusion. We did not mean to imply that the existence of this unique cell is the primary case for "antigenic switching"; that overall conclusion was based on the additional AP2IX-1 transfection experiment. This unique cell, however, does reveal some very important information, even though it is clearly an anomaly. It shows that within a population, rare variants can exist that are in completely different developmental states, in this case a parasite that has partially switched into the sexual stages of development normally seen only in the cat intestine. That is, from a single RH cell that lacks the expression of tachyzoite-specific SAG1, we identified expression of genes associated with sexual stages, including two AP2 transcription factors. We thus hypothesized that one or both of the AP2 transcription factors may play a causal role in regulating the expression of SAG1 and sexual stage genes. We showed experimental support in favor of this hypothesis with Figure 5 panels E-F; transfection of an AP2 IX-1 plasmid led to significant upregulation of sexual stage genes that we identified previously in the lone parasite cell, including SRS22C which was up-regulated by over 1000-fold. While AP2 IX-1 transfection led to roughly 3-fold reduction of SAG1 protein in 20 hours, it did not lead to the complete disappearance of SAG1 protein which may require longer treatment time due to the long half-life of this mRNA (Cleary et al., Nature Biotechnology 23;232-237, 2005). We acknowledge that single-cell RNA-seq measurement of the transfected parasites would provide a more precise measurement; however, we believe our claim on the causal role of AP2 IX-1 mediating the switching of SAG1 to SRS22C is substantiated by the provided data. Importantly, we do not make claims as to the cause that led to the expression of AP2 transcription factors and sexual-stage genes in the lone SAG1- parasite from our single parasite dataset.